# New insights into the environmental factors controlling the ground thermal regime across the Northern Hemisphere: a comparison between permafrost and non-permafrost areas

Olli Karjalainen[1], Miska Luoto[2], Juha Aalto[2,3], and Jan Hjort[1]

[1]Geography Research Unit, University of Oulu, FI-90014, Oulu, Finland
[2]Department of Geosciences and Geography, University of Helsinki, FI-00014, Helsinki, Finland
[3]Finnish Meteorological Institute, FI-00101, Helsinki, Finland

*Correspondence to:* Olli Karjalainen (olli.karjalainen@oulu.fi)

**Abstract.** The thermal state of permafrost affects Earth surface systems and human activity in the Arctic and has implications to global climate. Improved understanding of the local-scale variability in the global ground thermal regime is required to account for its sensitivity to changing climatic and geoecological conditions. Here, we statistically related observations of mean annual ground temperature (MAGT) and active-layer thickness (ALT) to high-resolution (~1 km$^2$) geospatial data of climatic and local environmental conditions across the Northern Hemisphere. The aim was to characterize the relative importance of key environmental factors and the magnitude and shape of their effects on MAGT and ALT. The multivariate models fitted well to both response variables with average $R^2$ values being ~0.94 and 0.78, respectively. Corresponding predictive performances in terms of root mean square error were ~1.31 °C and 87 cm. Freezing (FDD) and thawing (TDD) degree-days were key factors for MAGT inside and outside the permafrost domain with average effect sizes of 6.7 °C and 13.6 °C, respectively. Soil properties had marginal effects on MAGT (effect size = 0.4–0.7 °C). For ALT, rainfall (effect size = 181 cm) and solar radiation (161 cm) were most influential. Analysis of variable importance further underlined the dominance of climate for MAGT and highlighted the role of solar radiation for ALT. Most response shapes for MAGT that were ≤ 0 °C and ALT were non-linear and indicated thresholds for covariation. Most importantly, permafrost temperatures had a more complex relationship with air temperatures than non-frozen ground. Moreover, the observed warming effect of rainfall on MAGT$_{\leq 0 °C}$ reverted after reaching an optimum at ~250 mm, and that of snowfall started to level off at ~300–400 mm. It is suggested that the factors of large global variation (i.e. climate) suppressed the effects of local-scale factors (i.e. soil properties and vegetation) owing to the extensive study area and limited representation of soil organic matter. Our new insights into the factors affecting the ground thermal regime at a 1-km scale should improve future hemispheric-scale studies.

## 1 Introduction

In the face of changing Arctic, it is crucial to understand the mechanisms that drive the current geocryological dynamics of the region. Thaw of permafrost is expected to significantly attribute to hydrological and geoecological alterations in landscapes (Jorgenson et al., 2013; Liljedahl et al., 2016). In addition, greenhouse gas emissions from thawing permafrost soils have a potential to affect the global climate system (e.g. Grosse et al., 2016). Permafrost temperature and the depth of the overlying seasonally thawed layer, i.e. active layer, are key components of the ground thermal regime that govern various geomorphological and ecological processes (Frauenfeld et al., 2007; Aalto et al., 2017), as well as human activity in permafrost regions (Callaghan et al., 2011; Vincent et al., 2017; Hjort et al., 2018). Outside the permafrost domain, extensive regions undergo seasonal freezing, which in itself affects many aspects of natural and human activities (e.g. Shiklomanov, 2012; Westermann et al., 2015).

Climatic conditions account for large-scale spatial variation in mean annual ground temperature (MAGT) and active-layer thickness (ALT) (Bonnaventure and Lamoureux, 2013; Streletskiy et al., 2015; Westermann et al., 2015). However, from

regional to local scales, topography-induced solar radiation input (Etzelmüller, 2013) and intercepting layers of snow, vegetation, and soil conditions mediate their effect (e.g. Osterkamp, 2007; Fisher et al., 2016; Gruber et al., 2017; Aalto et al., 2018a; Zhang et al., 2018). Soils have different heat conductivities between frozen or thawed states, which can result in notable temperature differences between ground surface and top of permafrost, i.e. thermal offset (e.g. Smith and Riseborough, 1996).

Winter temperatures have been suggested to be most important for permafrost temperature (Smith and Riseborough, 1996; Etzelmüller et al., 2011), while ALT is essentially dependent on summer temperatures (Oelke et al., 2003; Melnikov et al., 2004; Luo et al., 2016). In wintertime, snow layer insulates the ground from cold air causing surface offset, i.e. ground is warmer than air (e.g. Aalto et al., 2018b; Zhang et al., 2018). Rainfall alters the thermal conductivity of near-surface layers through its control on, e.g., soil water balance (Smith and Riseborough, 1996; Callaghan et al., 2011; Marmy et al., 2013).

Arguably, the responsiveness of the hemispheric ground thermal regime to atmospheric forcing also depends on its initial thermal state. In permafrost conditions, temperature changes are lagged by the higher demand of energy for phase changes of water in the active layer (i.e. latent-heat exchange), whereas in temperate soils climate signal affects the ground thermal regime more directly (Romanovsky et al., 2010; Kurylyk et al., 2014). In addition to the effect of ground ice content on heat transfer, its development is an important geomorphic factor (e.g. Liljedahl et al., 2016).

Improved knowledge on hemispheric-scale permafrost dynamics is required to understand various geoecological interactions and feedbacks associated with warming Arctic (e.g. Wu et al., 2012; Grosse et al., 2016; Yi et al., 2018). Such information is useful for climate change assessments (Zhang et al., 2005, Smith et al., 2009), infrastructure design and maintenance, as well as for adaptation to changing conditions (Romanovsky et al., 2010, Streletskiy et al., 2015; Hjort et al., 2018). Physically based ground thermal models can account for various biogeophysical processes acting in vegetation, snow and soil layers (e.g.

Lawrence and Swenson, 2011) but are not applicable at high spatial resolutions over large areas owing to their tedious model parameterizations (Chadburn et al., 2017). For example, commonly used circumpolar 0.5° latitude/longitude resolution has been considered insufficient in characterizing spatial variation in soil properties and vegetation, thus leading to large mismatch between the simulations and observations (Park et al., 2013). Recently, Peng et al. (2018) assessed spatio-temporal long-term trends in circumpolar ALT with a large observational dataset stressing that ALT strongly depends on local topo-edaphic factors

(e.g. Harlan and Nixon, 1978) and that thorough analyses of environmental factors controlling ALT at varying scales are urgently required.

Here, we use a statistical modelling framework employing multiple algorithms from regression to machine learning to examine the factors contributing to the spatial variation in the hemispheric ground thermal regime in areas with and without permafrost. More specifically, we aim to (1) calibrate realistic models of the ground thermal conditions utilizing field observations of

MAGT and ALT (the response variables) and geospatial data on climatic and local conditions (the predictors) at 1-km resolution across the Northern Hemisphere land areas, and (2) examine the relative importance, magnitude of effect, and response shapes of environmental factors. The focus of this study is on MAGT and ALT in permafrost regions but the analyses are also performed for sites with MAGT above 0 °C to compare factor importances, effect sizes and response shapes between the thermal regimes.

**2 Methods**

**2.1 Study area and observational data**

We compiled MAGT and ALT observations from the period 2000–2014 over the Northern Hemisphere land areas north of the 30[th] parallel (Fig.1). To examine possible differences in the contribution of environmental factors between permafrost and non-permafrost conditions we used two separate MAGT datasets; observed MAGT at or below 0 °C, i.e. permafrost, ($MAGT_{\leq 0\,°C}$,

n = 469) and above 0 °C ($MAGT_{>0\,°C}$, n = 315). For each MAGT and ALT site, averages over the study period were then

calculated from available annual averages or suitable single measurements. The observations were standardized by requiring that MAGT was recorded at or near the depth of zero annual amplitude (ZAA) where annual temperature variation was less than 0.1 °C, and that ALT (n = 298) values represented the maximum thaw depth of a given year based on mechanical probing or derived from ground temperature measurements or thaw tubes (Brown et al., 2000; Aalto et al., 2018a). When ZAA depth was not reported or not retrievable from numeric data, we used the value at the depth of 15 m, where annual temperature fluctuation in most conditions is negligible (see French, 2007), although in thermally highly diffusive subsurface materials, such as bedrock, the depth can be greater (e.g. Throop et al., 2012). With some MAGT observations, ZAA depth was reportedly not reached but we chose to include these cases assuming that annual means calculated from year-round records from one or multiple years were representative of long-term thermal state. MAGT measured at less than two metres below the surface were excluded unless reported to be at the depth of ZAA.

The Global Terrestrial Network for Permafrost database (GTN-P, Biskaborn et al., 2015) was the principal constituent of our datasets (~60 % of MAGT and ~67 % of ALT observations). Additionally, data were gathered from open Internet databases (e.g. Roshydromet, meteo.ru; Natural Resources Canada, GEOSCAN database; National Geothermal Data System) and previous studies to cover a maximal range of climatological and environmental conditions (see Table S1 and S2 for sources)

A minimum geopositional location precisions of two decimal degrees (~1,110 m at the Equator) for MAGT and a commonly used arc minute (~1,800 m) for often less accurately geopositioned ALT sites were adopted both to ascertain adequate spatial match with geospatial data layers and to moderate the need to exclude lower precision observations. Nonetheless, almost 90 % of MAGT and more than two-thirds of ALT observations had a precision of at least three decimal degrees (~110 m at the Equator). Further exclusions were made when the ground thermal regime was evidently disturbed by recent forest fire, anthropogenic heat source, large water bodies or the effect of geothermal heat in temperature-depth curve (Jorgenson et al., 2010; Woo, 2012) as revealed by source data or cartographical examination of the site.

### 2.2 Predictor variables

Nine geospatial predictors representing climatic (air temperature and precipitation) and local (potential incident solar radiation, vegetation and soil properties) conditions at 30 arc-second spatial resolution were selected to examine their potential effects on MAGT and ALT at the hemispheric scale (e.g. Brown et al., 2000; French, 2007; Jorgenson et al., 2010; Bonnaventure and Lamoureux, 2013; Streletskiy et al., 2015). Climatic parameters were derived from the WorldClim dataset (Hijmans et al., 2005). The temporal coverage of WorldClim is 1950–2000, so we adjusted the data to match our study period of 2000–2014 using the Global Meteorological Forcing Dataset for land surface modelling (GMFD, Version 2, Sheffield et al., 2006) at a 0.5-degree resolution (see Aalto et al., 2018a). Monthly averages over this 15-year period were then used to derive the following climate parameters.

Previous studies have suggested that using indices representing the length or magnitude of thawing and freezing season could be more suitable than annual mean of air temperature (e.g. Zhang et al., 1997; Smith et al., 2009). Thus, thawing (TDD) and freezing (FDD) degree-days were determined as cumulative sums of mean monthly air temperatures above and below 0 °C, respectively. Frauenfeld et al. (2007) showed that their use instead of daily temperatures accounted for less than 5 % error for most high-latitude land areas. Since available global data on snow thickness or snow-water equivalency have relatively coarse spatial resolutions (Bokhorst et al., 2016), we examined the snow cover's contribution indirectly using derivatives of the climate data. We estimated annual precipitation as water droplets (hereafter rainfall) or snow particles (hereafter snowfall) by summing up precipitation (mm) for months with mean monthly temperature below and above 0 °C, respectively (Zhang et al., 2003).

MODIS Terra-based normalized difference vegetation indices (NDVI, Didan, 2015) at a 1-km resolution were used to assess the amount of photosynthetic vegetation. We averaged monthly summertime (June to August) NDVI values over the study

period of 2000–2014 and screened for only high-quality pixels based on the MODIS pixel reliability attribute. Potential incident solar radiation, computed after McCune and Keon (2002, Equation 2, p. 605) utilizing slope angle and aspect, along with latitude, was used to estimate the potential incident solar radiation (SolarRad, W cm$^{-1}$ a$^{-1}$) that affects the energy balance of the ground thermal regime (e.g. Hasler et al., 2015; Streletskiy et al., 2015). Ground temperatures at sites with thin or no overlying unconsolidated sediments above bedrock have been shown to be more closely coupled with air temperatures than those with thick overburden and associated latent heat effects and lower thermal diffusivity (e.g. Throop et al., 2012). The effects of overburden thickness, however, could not be assessed due to the lack of suitable global fine-resolution data. To account for the thermal offset dictated by soil properties (e.g., Smith and Riseborough 1996, 2002; Kurylyk et al., 2014) we extracted soil organic carbon content (SOC, g kg$^{-1}$), and fractions of coarse (CoarseSed, > 2 mm) and fine sediments (FineSed, ≤ 50 μm) for 0–200 cm subsurface from SoilGrids database (Hengl et al., 2017).

## 2.3 Statistical modelling

### 2.3.1 Calibration of MAGT and ALT models

We used four statistical techniques, namely generalized linear modelling (GLM, McCullagh and Nelder, 1989), generalized additive modelling (GAM, Hastie and Tibshirani, 1990), and regression-tree based machine-learning methods generalized boosting method (GBM, Friedman et al., 2000) and random forest (RF, Breiman, 2001) to calibrate MAGT and ALT models by using the nine geospatial predictors. Multi-model framework was adopted to control for uncertainties related to the choice of modeling algorithm (e.g. Marmion et al., 2009). GLM is an extension of linear regression capable of handling non-linear relationships with an adjustable link function between the response and explanatory variables. The GLM models were fitted including quadratic terms for each predictor. In GAM, alongside linear and polynomial terms, smoothing splines can be applied for more flexible handling of non-linear relationships. For smoothing spline, a maximum of three degrees of freedom were specified, which was further optimized by the model fitting function. To examine the direction and possible non-linearity of the relationship between predictors and responses, we used GAM to plot model-based response curves. The curves show smoothed fit between response and a predictor while all other predictors are fixed at their average (Hjort and Luoto, 2011). Both GLM and GAM were fitted without interactions between predictors using a Gaussian error distribution with an identity link function.

GBM was specified with the following parameters: number of trees = 3,000, interaction depth = 6, shrinkage = 0.001. Bagging fraction was set to 0.75 to select a random subset of 75 % of the observations at each step, without replacement. As for RF, 500 trees, each with a minimum node size of five were grown. The final prediction is the average of individual tree predictions. Both GBM and RF automatically consider interaction effects between predictors (Friedman et al., 2000). All statistical analyses were executed in R (R Core team, 2015) using the base and auxiliary R packages; *mgcv* (Wood, 2011) for GAM, *dismo* (Hijmans et al., 2016) for GBM, and *randomForest* for RF (Liaw and Wiener, 2002).

### 2.3.2 Model evaluation

To evaluate the models, we split the response data randomly into calibration (70 % of the observations) and evaluation (30 %) datasets (Heikkinen et al., 2006). This was repeated 100 times, at each step fitting models with the calibration data and then using them to predict to both the calibration and evaluation datasets. Model performance was assessed with adjusted coefficient of determination ($R^2$) and root mean square error (RMSE) between observed and predicted values in these datasets.

### 2.3.3 Variable importance computation

A measure of variable importance was computed to determine the relative importance of each predictor to the models´ predictive performance (Breiman, 2001). In the computation, each modelling technique was first used to fit models with the MAGT and ALT datasets using all the nine predictors. The variable importance was then computed based on Pearson's correlation between predictions from two models produced with the fitted model; one with unchanged variables, and another where the values of one variable were randomized while others remained intact. In the procedure, each predictor was randomized in successive model runs. The measure of variable importance was computed as follows:

Variable importance = $1 - \text{corr}(\text{Prediction}_{\text{intact variables}}, \text{Prediction}_{\text{one variable randomized}})$ (1)

On a range from 0 to 1, high variable importance value, i.e. high individual contribution to MAGT or ALT, was returned when any randomized predictor had a substantial impact on the model's predictive performance, and consequently resulted low correlation with predictions from the model with intact variables (Thuiller et al., 2009). Each modelling method was run 100 times for each response with each predictor shuffled separately. For each run, different subsample from the original data was randomly bootstrapped with replacement.

### 2.3.4 Effect size statistics

Effect sizes for each predictor were determined based on the range between the predicted minimum and maximum MAGT and ALT values over the observation data while controlling for the influence of other predictors by fixing them at their mean values (see Nakagawa and Cuthill, 2007). The procedure was repeated with each dataset and modelling method.

### 3 Results

MAGT in permafrost conditions was on average −3.1 °C while the minimum was −15.5 °C. $\text{MAGT}_{>0\,°C}$ had an average of 8.0 °C and a maximum of 23.2 °C. ALT had an average of 141 cm and ranged from 23 to 733 cm. The extreme values, apart from the ALT maximum, were based on one year of measurements. Pairwise correlations and the scatter plots revealed a strong association between MAGT and air temperature (see Smith and Burgess, 2000; Smith and Riseborough, 2002; Throop et al., 2012), especially for $\text{MAGT}_{>0\,°C}$ (Fig. 2a–b, d). In contrast to MAGT, ALT was not significantly correlated with TDD, but had stronger associations with soil properties (Fig. 2c). Coarse sediments and SOC, especially, were important and showed clear, yet non-linear, responses to ALT. Statistical descriptives of the predictors in respective datasets are presented in Fig. S1.

### 3.1 Model performance

$\text{MAGT}_{>0\,°C}$ models had the highest $R^2$ values between predicted and observed MAGT (Table 1). In permafrost conditions, all the models had high $R^2$ values for MAGT, whereas in case of ALT between-model variation was large and $R^2$ on average lower. A decrease in the fit was identified when predicting ALT to evaluation datasets, especially with GBM and RF, whereas MAGT models retained their high performance. On average, RMSEs were low (~1 °C) in $\text{MAGT}_{≤0\,°C}$ and $\text{MAGT}_{>0\,°C}$ calibration datasets. When predicted over evaluation datasets, the average increased slightly more in non-permafrost conditions. A similar increase of 40 % was documented with ALT. For each response, GBM and RF had lower RMSEs (i.e. higher predictive performance) than GLM and GAM, but also larger change between calibration and evaluation datasets, indicating that GLM and GAM produced more robust predictions.

### 3.2 Relative importance of individual predictors

FDD and TDD were the most important factors affecting MAGT; FDD (variable importance score = 0.27) where permafrost was present, TDD (0.53) in non-permafrost conditions (Fig. 3a–b). Precipitation predictors, especially rainfall, had a moderate importance (0.10) on $\text{MAGT}_{≤0\,°C}$ but were marginal when permafrost was not present (0.01). Climatic factors were followed

by solar radiation (0.02, both MAGT datasets) and finally by NDVI and soil properties with minimal importance (each $\leq 0.01$). The importance of both rainfall and snowfall was higher in permafrost conditions.

Solar radiation was the most important predictor (0.37) explaining variation in ALT (Fig. 3c). Rainfall had second highest importance (0.05) followed by soil properties SOC (0.04) and coarse sediments (0.03). The remaining climate variables (snowfall, TDD and FDD) had low importance scores that were comparable to those of NDVI (each 0.01–0.02).

### 3.3 Effect size of individual predictors

FDD had the highest individual effect size of 6.7 °C averaged over the four methods in case of $MAGT_{\leq 0 °C}$, whereas in $MAGT_{>0 °C}$ dataset TDD accounted for a dominant 13.6 °C effect (Table 2). Precipitation had the second highest effect, albeit snowfall was less effective in non-permafrost conditions. Considering the remaining predictors, clear differences were observed in cases of SOC and NDVI, both higher in $MAGT_{>0 °C}$ dataset. In case of ALT, solar radiation retained a central role while rainfall exerted the greatest average effect (181 cm) despite large between-model variation. In contrast to variable importance results (Fig. 3c), snowfall had a larger average effect than coarse sediments and SOC, both of which nevertheless had a considerable effect.

### 3.4 Response shapes

A varying degree of non-linearity was visible in the responses between $MAGT_{\leq 0 °C}$ and the key predictors, whereas in case of $MAGT_{>0 °C}$ the responses were more often linear (Fig. 4a–b). Moreover, thresholds for covariation were visible in permafrost conditions. For example, a flat relationship with MAGT and FDD turned into a negative as FDD increased, and rainfall had a unimodal (i.e. humped) curve depicting an optimum level in the relationship with MAGT. Below the optimum rainfall had a warming effect, and above a cooling effect occurred. ALT had non-linear response shapes with all the predictors in Fig. 4c. Solar radiation had a highly non-linear unimodal curve with at an optimum located around 0.7 W cm$^{-1}$ a$^{-1}$. Response curves for the remaining predictors with smaller contribution were predominantly linear and flat indicating relatively modest effects (Fig. S2).

## 4 Discussion

### 4.1 Factors affecting MAGT and ALT

Our results are in line with previous understanding that climatic conditions are the primary factors affecting the long-term averages of MAGT across the Northern Hemisphere at 1-km resolution but also indicate that the magnitude and shape of the effects of TDD and FDD on MAGT are dependent on permafrost presence or absence. As anticipated, FDD has higher influence on MAGT in permafrost conditions where strong freezing occurs (e.g. Smith and Riseborough, 1996). At sites without permafrost, TDD has a nearly linear dominant (Fig. 4b) effect, which is suggested to be mostly attributed to the lack of the buffering effect of the freeze-thaw processes and latent-heat exchange in the active layer (e.g. Osterkamp, 2007), and to the absence of seasonal snow cover in the warmest parts of the study region. In permafrost conditions, the warming effect of TDD and especially the cooling effect of FDD on MAGT show flattening in response shapes where MAGT is close to 0 °C owing to the latent-heat effects associated with thawing and freezing of water in the active layer (Fig. 4a). These findings suggest that near-zero permafrost temperatures are less responsive to air temperatures than cold permafrost or non-frozen ground. Observed non-linearities clearly illustrate the importance of using modelling techniques and parameterizations capable of recognizing the points where system behavior changes. In Earth surface system studies, non-linear responses are often more common than linear ones, and assuming linear relationships against theoretical or empirical evidence may result in biased outcomes (Hjort and Luoto, 2011).

The minimal effect of TDD on ALT contradicts with the documented strong regional scale (spatio)temporal connection (e.g. Zhang et al., 1997; Oelke et al., 2003; Frauenfeld et al., 2004; Melnikov et al., 2004; Yi et al., 2018). According to our results, the spatial linkage is more elusive at a broader scale and could be attributed to the great hemispheric variation in ALT. The majority of high-Arctic sites locate on low-lying tundra overlaid by mineral and organic soil layers, whereas mid-latitude sites predominantly locate in mountains (the European Alps, central Asian mountain ranges) with thin soils and thermally diffusive bedrock. This difference partly explains generally small and large ALT within the respective regions notwithstanding that they can have similar average climatic conditions (e.g. TDD, see Fig. 2d). Moreover, large inconsistencies between observed ALT and climate-warming trends have been documented (e.g. Wu et al., 2012; Gangodagamage et al., 2014). Although temporal dynamics of ALT are beyond our analyses, our findings suggest that thaw depth and air temperatures are, to a degree, decoupled by local conditions.

Recent warming trends in the atmosphere (Guo et al., 2017) are already well visible in circumpolar permafrost temperature observations (Romanovsky et al., 2017; Biskaborn et al., 2019) implying that the permafrost system will remain dynamic in future's changing climate. Warmer air temperatures will occur mostly during winters (AMAP, 2017; Guo et al., 2017), which, given the presented high contribution of FDD on MAGT, suggests that changes are foreseeable. Other climatic factors, however, bear significance. Biskaborn et al. (2019), for example, reported a ground warming in the discontinuous permafrost zone between 2007 and 2016 due to increase in snow thickness even though no significant change in air temperature during a similar period occurred. Projected warmer winters can also affect ALT through changing snow conditions and subsequent changes in hydrology and vegetation (Park et al., 2013; Atchley et al., 2016; Peng et al., 2018).

Our results highlight the notable role of rainfall on both MAGT and ALT (Peng et al., 2018; Zhang et al., 2018). Projected greater proportion of rainfall (e.g. AMAP, 2017; Bintanja and Andry, 2017) potentially has a direct effect on the ground thermal regime through its influence on latent heat exchange (Westermann et al., 2011), and convective warming during spring (Kane et al., 2001) and summertime (Melnikov et al., 2004; Marmy et al., 2013). However, abundant summer rains arguably also cool the ground surface through increased evaporation and heat capacity, and thus limit the heat conduction into the ground (Zhang et al., 1997, 2005; Frauenfeld et al., 2004; Park et al., 2013). In permafrost conditions, the warming effect of rainfall for MAGT is indeed found, but only up to ~250 mm above which it reverts to a cooling (Fig. 4a). The response with ALT, in turn, is relatively flat up to a point where abundant rainfall (> ~500 mm) leads to a strong deepening of ALT. It should be noted that due to the small amount of $MAGT_{\leq 0\,°C}$ and ALT sites with rainfall above 400–500 mm (Fig. S1), the confidence intervals for the response curves are relatively large demonstrating high amount of uncertainty.

The dominant contribution of rainfall over snowfall observed here contradicts with some previous regional scale studies (e.g., Zhang et al., 2003, 2005). However, the elevated effect of snowfall on $MAGT_{\leq 0\,°C}$ (effect size of 2.3 °C compared to 0.8 °C in non-permafrost conditions) underlines the role of snow cover's control over the thermal regime of permafrost-affected ground. Similarly, Zhang et al. (2018) found that the offset between air and surface temperatures was weaker in temperate regions (mean annual air temperature > 0 °C) than in low-Arctic and boreal permafrost regions, although also high-Arctic had small surface offsets owing to small amount of snow. For permafrost conditions, non-linear response shape indicates that the warming effect of snow starts to level off at around 300–400 mm. Previous studies have shown that the insulating effect levels off when snow reaches certain depth, e.g., about 400 mm (Zhang, 2005). Despite the complexity involved in the role of snow conditions (e.g. Fiddes et al., 2015; Aalto et al., 2018b), thick snow cover has been shown to increase also ALT at site (Atchley et al., 2016), regional (Zhang et al., 1997; Frauenfeld et al., 2004) and circumpolar scale (Park et al., 2013). Here, active-layer thickening is visible only after relatively high snowfall values (~700 mm). However, this effect is based on a limited set of ALT sites (less than 10% of the ALT sites had snowfall exceeding 300 mm) and is therefore uncertain.

Incoming solar energy can be considered central for soil thawing (see Biskaborn et al., 2015), but the high contribution of solar radiation on ALT stands out. Arguably, the effect is emphasized because ALT observation sites in cold permafrost conditions are mostly sparse in vegetation and lack tree canopy (Zhang et al., 2003; Biskaborn et al., 2015). Moreover, most of the ALT sites have been established on flat terrain (Biskaborn et al., 2015), meaning that local topographic shading is less significant. Thus, ALT is suggested to follow poleward decrease in solar radiation and associated shorter thaw seasons (see Luo et al., 2016). The weaker association of solar radiation with MAGT suggests that its direct effect is limited to the near-surface permafrost, i.e. intensified thawing during thawing seasons, and that the influence to deeper temperatures is more indirect and associated with the relationship between annual solar radiation and air temperatures. Moreover, given that MAGT sites are usually located in more topographically heterogeneous terrain than ALT sites, the local exposure to solar radiation is suggested to be more important than the latitudinal trend (e.g. Romanovsky et al., 2010). These suggestions are corroborated by the response shapes (Fig. 4a, c). The response curve of $MAGT_{\leq 0\,°C}$ is flat until a relatively high solar radiation of ~0.7 W cm$^{-1}$ a$^{-1}$, whereas a strong increase in ALT terminates at these values and reverts to thinning. The end of the response function presumably reflects the latitudinal gradient, i.e., the sites in the Tibetan Plateau with high solar radiation but relatively small ALT.

The weak connection between TDD and ALT is additionally explained by soil factors that influence the heat transfer between the lower atmosphere and the ground (Smith et al., 2009). According to the response shapes from GAM, coarse sediments increase ALT when enough prevalent (~25 % fraction) in the soils. The effect of soil texture on ALT has been implied to occur largely through its effects on hydrological conditions (Zhang et al., 2003; Yin et al., 2017) and conductivity (Callaghan et al., 2011). More efficient water transfer in coarse-grained material could impose convective heat into soils during the thawing season or promote latent-heat effect during the freeze-up, which both contribute to deeper thaw (see Romanovsky and Osterkamp, 2000; Frauenfeld et al., 2004). Thermal insulation by soil organic layers has been demonstrated to effectively decouple air-permafrost connection resulting in thinner active layer and lower soil temperatures (e.g. Johnson et al., 2013; Atchley et al., 2016). The GAM response shape illustrates a thinning of ALT with increasing SOC until ~150 g kg$^{-1}$, after which additional organic carbon does not attribute to enhanced insulation. It should be noted that the used variable depicts SOC in fine earth fraction and does not explicitly address incompletely decomposed or fresh organic matter, which are one of the central components of the thermal offset. However, suitable gridded data on soil organic matter content are not available, and physical fractionation of SOC has been commonly used as its correlative proxy owing to more straightforward measurement procedures (Bailey et al., 2017).

NDVI has a small contribution on ALT and MAGT in permafrost conditions, but outside the permafrost region it has a moderate linear cooling effect. The low contribution of NDVI in permafrost conditions could be attributed to the intra- and inter-seasonal differences in the effects of different vegetation canopy configurations that can have similar index values. For example, in wintertime, tall shrubcanopy traps snow and thereby enhances insulation of the ground (Morse et al., 2012), whereas taller tree canopies of evergreen boreal forestsintercept snow and allow more heat loss from the ground in winter, and in summer their shading cools the ground surface (Lawrence and Swenson, 2011; Fisher et al., 2016).

## 4.2 Uncertainties

Large-scale scrutinization of factors affecting ground thermal dynamics is often hindered by data deficiencies or unavailability. More precisely, many data lack adequate spatial or temporal accuracy, geographical consistency, methodological robustness or thematic detail (Bartsch et al., 2016; Chadburn et al., 2017). Some of these shortcomings are exacerbated in remote permafrost regions with low-density observational networks of, e.g., climatic parameters (Hijmans et al., 2005) or soil profiles (Hengl et al., 2017). The fine-scale spatial variability of ALT and MAGT called for a high spatial resolution data to assess the local factors that mediate the atmospheric forcing. Here, the availability of geospatial data largely determined the resolution

of 30 arc seconds, which could be considered the highest currently attainable resolution at a near-global scale. While not adequate to account for all potential sources of sub-grid spatial heterogeneity in, e.g. microclimatic conditions, especially in topographically complex conditions (Fiddes et al., 2015; Aalto et al., 2018b; Yi et al., 2018), the implemented resolution is a step forward in making a distinction in between-site conditions and revealing local relationships relevant at the hemispheric-scale.

In general, the sensitivity of MAGT to the climatic parameters along with the minimal role of soil and vegetation properties suggests that future MAGT is more feasible to predict than ALT, even without addressing, for example, future vegetation or soil organic carbon content, whose response to climate change is extremely challenging to project (Jorgenson et al., 2013). This is incongruent with previous studies showing the high importance of soil properties for MAGT (e.g. Zhang et al., 2003; Throop et al., 2012). The discrepancies are argued to be partly attributed to the hemispheric study extent; large spatial variation in climatic parameters is suggested to have suppressed the effect of soil and vegetation properties locally. It is also possible that the used SOC data could not fully address the thermal offset albeit ALT modelling showed a realistic response shape and a moderately strong effect. Another soil property likely affecting MAGT and ALT is the thickness of overburden materials above bedrock. Suitable data, however, were not available to scrutinize this. For example, the depth to bedrock predictions in the SoilGrids data (Hengl et al., 2017) were not sufficient for assessing realistic responses because one of the measures considers the overburden thickness (depth to R horizon, i.e. intact regolith) only within the first two metres below surface. While another measure in the SoilGrids data covers the range of measured MAGT depths, it has an RMSE of over 800 cm making it simply too imprecise. However, the effects of soil properties on MAGT have been shown to be statistically significant also when predicting future hemispheric ground thermal conditions (Aalto et al., 2018a), and should thus be accurately reproduced by geospatial data.

Given the pronounced role of precipitation, more direct information on fine-scale soil moisture conditions controlled by local soil and land surface properties (see Kemppinen et al., 2018), as well as more comprehensive and finer resolution data on global snow thickness are required for improved ground thermal regime modelling. Fine-scale biophysical factors affecting drainage conditions and distribution of wind-drifted snow (e.g. vegetation and small topographic depressions) are largely averaged-out and cannot be accounted for at 1-km resolution.

Although the main factors were identified as important and effective by each modelling technique, notable inter-modal variability suggested that using only one method could have led to disputable results. A multi-model approach was in this sense safer, although not all the methods may have worked optimally with the present observational and environmental data owing to their different abilities to handle collinearity, spatial autocorrelation or non-linearity. For example, interactions between variables were not included in regression-based modelling (GLM and GAM), while being intrinsically considered by tree-based methods (GBM and RF) (Friedman et al., 2000). Differences such as this could have attributed to the dissimilar performances of the models; GBM and RF were overall less stable when comparing $R^2$ and RMSE values between the observed and predicted values in calibration and evaluation settings. In the effect size analysis of ALT, GLM and GAM were possibly sensitive to the significantly higher rainfall and snowfall values at the few mountain sites (mainly in the European Alps). This might have caused the large effect sizes, partly incongruent with the variable importance results. Subsequently, the confidence intervals for these response curves were large, although uncertainties were small for most of the other predictors indicating that the observational data sufficiently covered the environmental gradients (Hjort and Luoto, 2011). We suggest that in addition to spatially and temporally high-quality precipitation data, more ALT monitoring sites from alpine environments are needed to more realistically assess the ALT-precipitation relationship.

## 5 Conclusions

We statistically related observations of MAGT and ALT to high-resolution (~1 km$^2$) geospatial data of climatic and local environmental conditions to explore the factors affecting the ground thermal regime of permafrost and non-permafrost areas across the Northern Hemisphere. Our modelling framework efficiently captured the multi-variate nature of ground thermal regime and highlighted the differences between the relative importance and effect size of climatic factors on MAGT inside and outside the permafrost domain. In permafrost conditions, climate was paramount and soil properties showed marginal role for MAGT, while precipitation factors and topography-controlled solar radiation were emphasized for ALT. Where permafrost was not present, precipitation was less influential and MAGT was predominantly controlled by air temperatures above 0 °C. We suggest that the large variation in climate predictors suppressed some of the local effect of soil properties and vegetation. The relatively minor role of soil properties (especially organic carbon content) on MAGT and ALT may have additionally stemmed from the lack of global data with high local accuracy

The results also revealed distinct non-linear relationships and thresholds between the ground thermal regime and environmental factors, especially in permafrost-affected regions. At sites without permafrost, responses were more often linear. Based on the response shapes, permafrost temperatures were less responsive to air temperatures than non-frozen ground. We also found that the warming effects of rainfall on MAGT$_{\leq 0\,°C}$ reverted after reaching an optimal level, and that of snowfall started to level off at about 350 mm. Even though these are hemispheric-scale estimates, they show that consideration of non-linear responses is vital when studying the thermal regime of permafrost-affected ground and addressing the impacts of changing air temperature or precipitation regime.

In addition to providing detailed characterizations of the key contributing factors at hemispheric scale, we conclude that multi-variate modelling frameworks capable of addressing the inherent non-linearity in Earth surface systems and employing high-resolution geospatial data will be valuable for, e.g., assessing permafrost degradation from local to global scale. We suggest that comparable broad-scale assessments should be performed that would further discriminate continuous, discontinuous, and less extensive permafrost zones or geoecologically distinct regions. This would facilitate the understanding of region-specific aspects of climate-permafrost relation, which is a prerequisite for the development of accurate and locally applicable future ground thermal projections.

### Author contribution

OK, ML and JH developed the original idea. OK led the compilation of observational data and geospatial data processing with contributions from all the authors. ML, OK and JA performed the statistical analyses. OK wrote the manuscript with contributions from all the authors.

*Acknowledgements.* This study was funded by the Academy of Finland (grants 285040,286950 and 315519). We wish to thank two anonymous reviewers and the Editor for their constructive comments that substantially improved the manuscript.

### Competing interests

The authors declare that they have no conflict of interest.

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

**Table 1: Adjusted coefficient of determination ($R^2$) and root mean square error (RMSE) between observed and predicted mean annual ground temperature (MAGT) and active-layer thickness (ALT) in calibration and evaluation (in brackets) datasets averaged over 100 permutations.**

| Method | $MAGT_{\leq 0\,°C}$ $R^2$ | RMSE (°C) | $MAGT_{>0\,°C}$ $R^2$ | RMSE (°C) | ALT $R^2$ | RMSE (cm) |
|---|---|---|---|---|---|---|
| GLM | 0.86 (0.83) | 1.24 (1.33) | 0.95 (0.92) | 1.20 (1.44) | 0.65 (0.50) | 80 (93) |
| GAM | 0.88 (0.84) | 1.17 (1.29) | 0.95 (0.92) | 1.18 (1.37) | 0.70 (0.54) | 74 (89) |
| GBM | 0.93 (0.86) | 0.88 (1.22) | 0.97 (0.92) | 0.91 (1.37) | 0.84 (0.59) | 55 (84) |
| RF | 0.98 (0.87) | 0.51 (1.17) | 0.99 (0.93) | 0.55 (1.27) | 0.93 (0.62) | 36 (82) |
| Average | 0.91 (0.85) | 0.95 (1.25) | 0.96 (0.92) | 0.96 (1.36) | 0.78 (0.56) | 61 (87) |

GLM = generalized linear modelling, GAM = generalized additive modelling, GBM = generalized boosting method and RF = random forest.

**Table 2: The effect size of individual predictors and their four-model averages (see Sect. 2.2 for abbreviations) in the original scale of the responses, °C for (mean annual ground temperature) MAGT and cm for active-layer thickness (ALT).**

| | $MAGT_{\leq 0\,°c}$ (°C) GLM | GAM | GBM | RF | Avg | $MAGT_{>0\,°c}$ (°C) GLM | GAM | GBM | RF | Avg | ALT (cm) GLM | GAM | GBM | RF | Avg |
|---|---|---|---|---|---|---|---|---|---|---|---|---|---|---|---|
| **FDD** | 8.6 | 10.7 | 4.3 | 3.2 | 6.7 | 3.8 | 4.3 | 2.6 | 2.8 | 3.4 | 117 | 86 | 15 | 36 | 64 |
| **TDD** | 7.1 | 6.6 | 2.4 | 2.8 | 4.7 | 19.1 | 19.5 | 9.0 | 6.6 | 13.6 | 30 | 23 | 19 | 31 | 26 |
| **Rainfall** | 1.6 | 2.6 | 4.3 | 3.0 | 2.9 | 4.8 | 3.6 | 0.2 | 0.7 | 2.3 | 372 | 249 | 28 | 74 | 181 |
| **Snowfall** | 4.4 | 4.4 | 0.1 | 0.2 | 2.3 | 0.8 | 1.4 | 0.3 | 0.5 | 0.8 | 195 | 146 | 44 | 94 | 120 |
| **SolarRad** | 2.6 | 2.5 | 0.2 | 0.3 | 1.4 | 2.0 | 2.3 | 0.9 | 1.6 | 1.7 | 135 | 193 | 178 | 139 | 161 |
| **CoarseSed** | 0.8 | 1.8 | 0.1 | 0.2 | 0.7 | 0.6 | 2.6 | 0.1 | 0.3 | 0.9 | 129 | 137 | 69 | 65 | 100 |
| **FineSed** | 0.5 | 0.7 | 0.2 | 0.4 | 0.4 | 0.6 | 0.7 | 0.1 | 0.1 | 0.4 | 17 | 20 | 7 | 9 | 13 |
| **SOC** | 0.5 | 0.4 | 0.3 | 0.8 | 0.5 | 1.7 | 1.4 | 0.1 | 0.6 | 0.9 | 121 | 129 | 30 | 28 | 77 |
| **NDVI** | 0.4 | 0.3 | 0.1 | 0.8 | 0.4 | 2.6 | 2.3 | 0.2 | 0.1 | 1.3 | 68 | 36 | 15 | 34 | 38 |

The values are shaded with increasing blue ($MAGT_{\leq 0\,°C}$), red ($MAGT_{>0\,°C}$) and yellow (ALT) hues relative to the magnitude of the effect. GLM = generalized linear modelling, GAM = generalized additive modelling, GBM = generalized boosting method and RF = random forest. See Sect. 2.2 for predictor abbreviations.

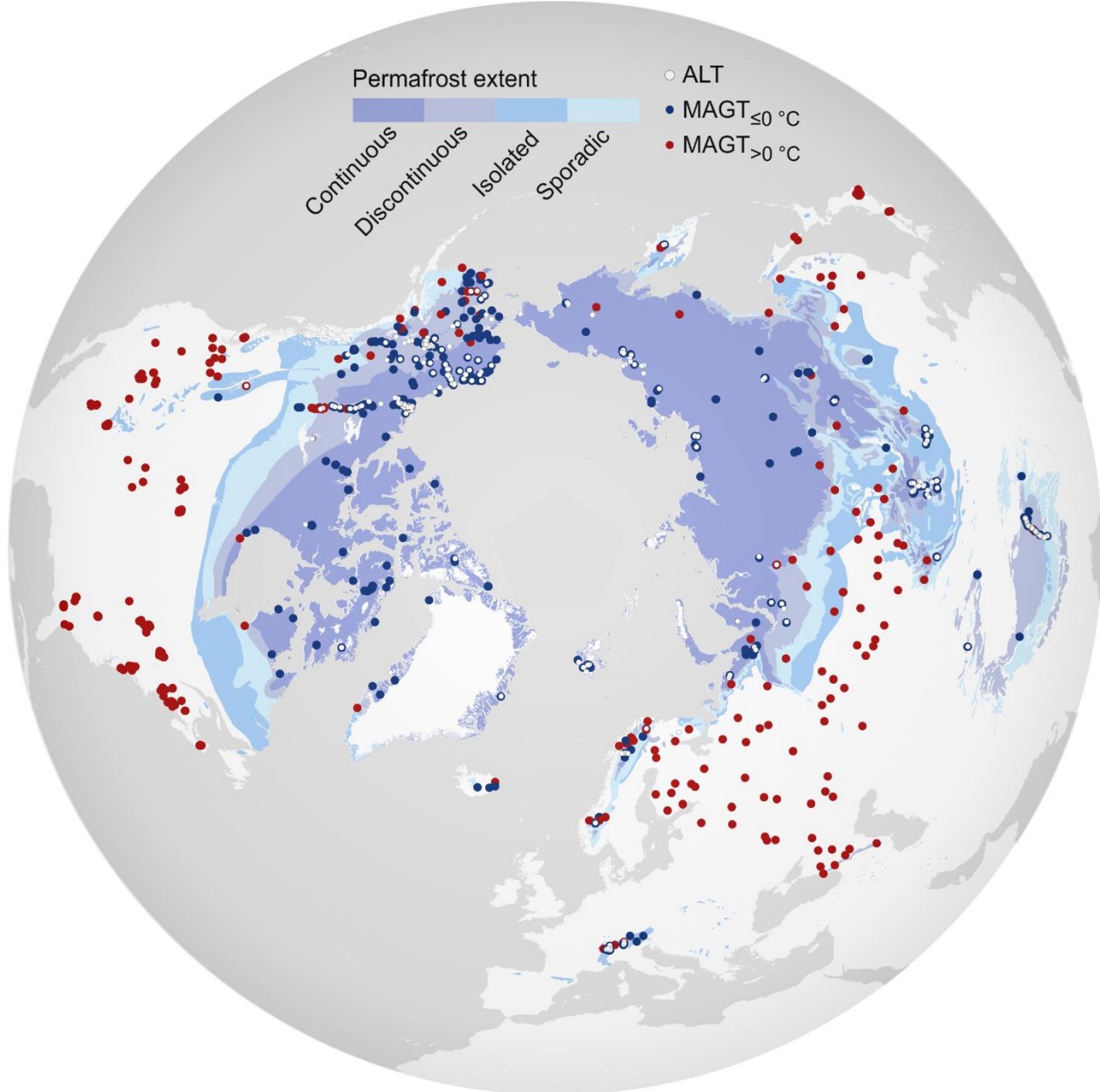


**Figure 1: The observational network of the mean annual ground temperature (MAGT) and active-layer thickness (ALT) measurement sites across the Northern Hemisphere that were used in this study. Blue symbols indicate the locations of boreholes where MAGT (averaged over the period 2000–2014) was at or below 0 °C and red symbols for those above 0 °C. White symbols depict the ALT measurements sites. The underlying permafrost zonation is from Brown et al. (2002).**


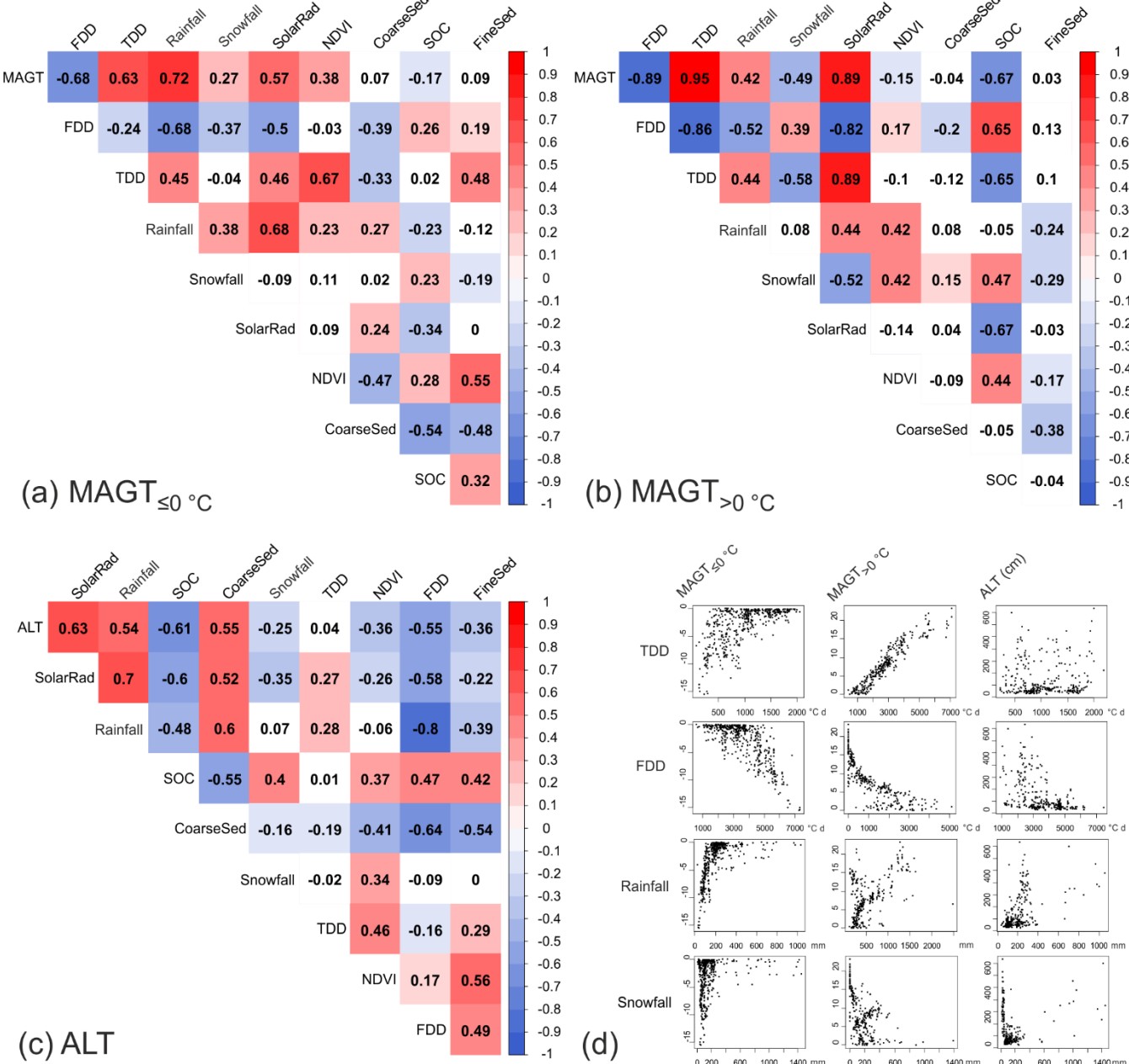

**Figure 2: Spearman rank-order correlations between the predictor variables (see Sect. 2.2 for abbreviations) and MAGT$_{\leq 0\,°C}$ (mean annual ground temperature) (a), MAGT$_{>0\,°C}$ (b) and ALT (active-layer thickness) (c). Red hue stands for positive correlations, blue for negative, and white indicates non-significant (p > 0.01) correlations. Panel (d) shows MAGT and ALT observations plotted against the climatic predictors.**

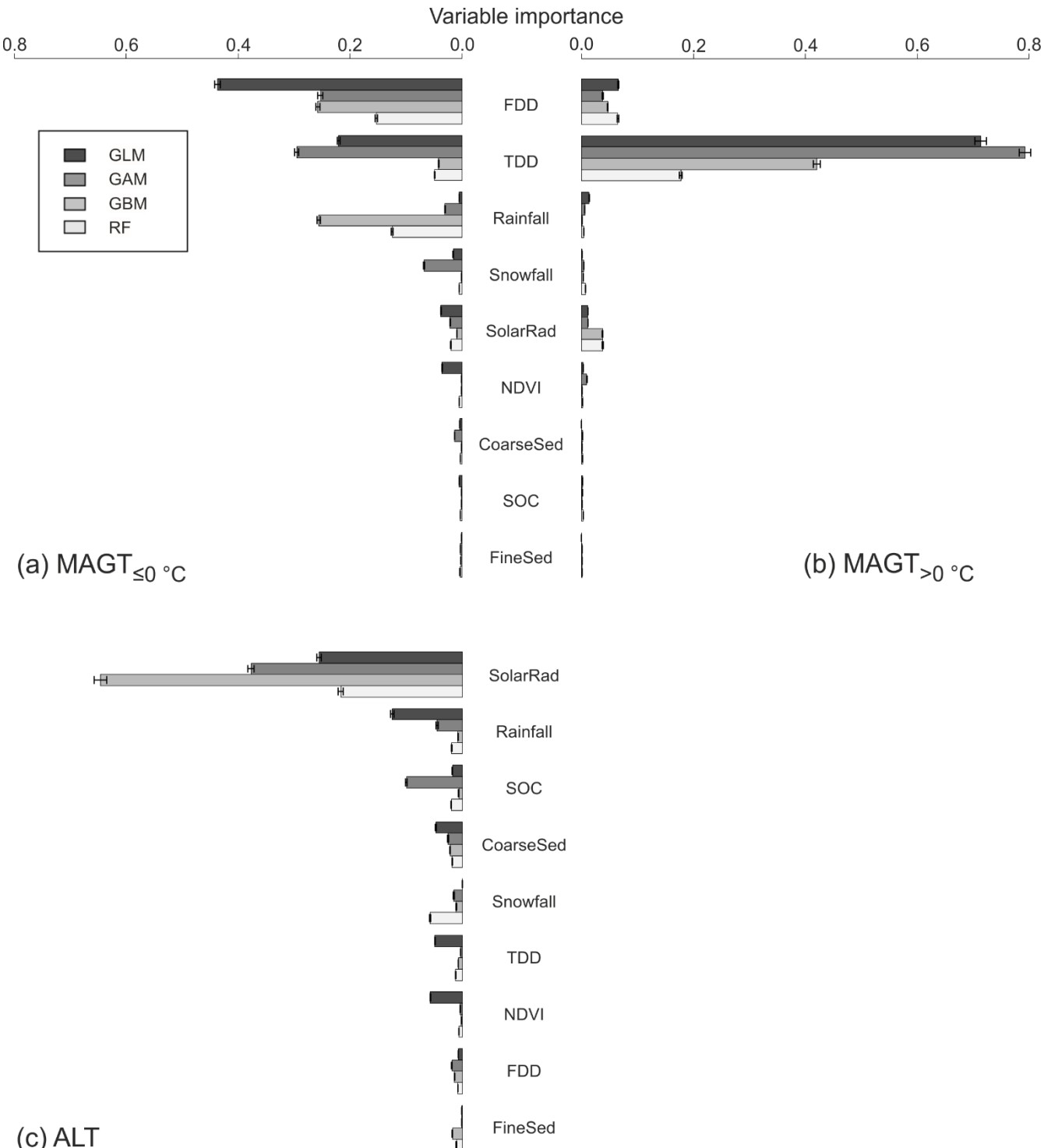

**Figure 3: Variable importance values in MAGT$_{\leq 0\ °C}$ (mean annual ground temperature less than or equal to 0 °C) (a) and MAGT$_{>0\ °C}$ (mean annual ground temperature greater than 0 °C) (b) datasets arranged in the descending order of four-model average in MAGT$_{\leq 0\ °C}$ conditions, and for ALT (active-layer thickness) (c), arranged likewise based on ALT results. The whiskers depict 95 % confidence intervals (over 100 bootstrapping rounds). GLM = generalized linear modelling, GAM = generalized additive modelling, GBM = generalized boosting method and RF = random forest. See Sect. 2.2 for predictor abbreviations.**

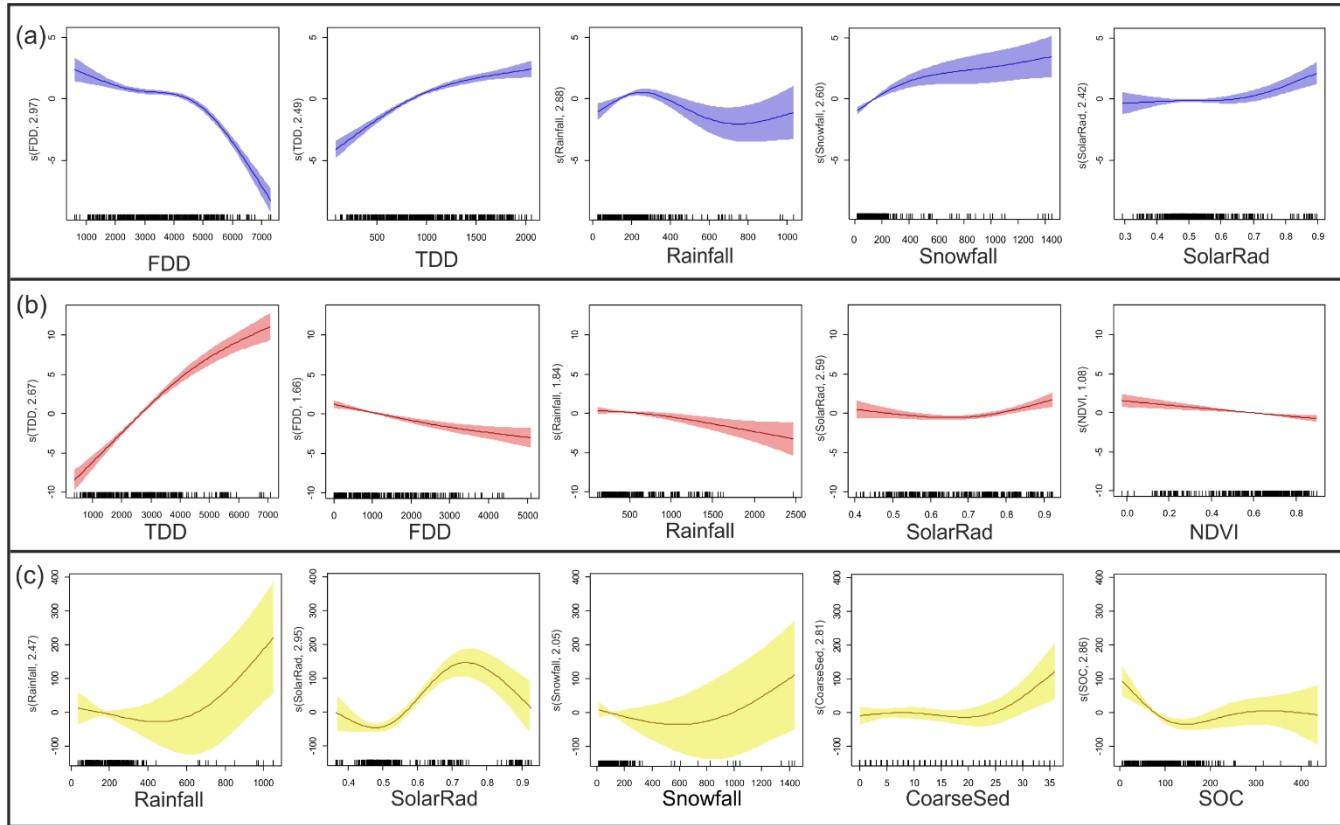


**Figure 4: Response shapes of the five predictors with most contribution in MAGT$_{\leq 0\ °C}$ (mean annual ground temperature less than or equal to 0 °C, blue curves) (a), MAGT$_{>0\ °C}$ (mean annual ground temperature greater than 0 °C, red curves) (b), and ALT (active-layer thickness, yellow curves) (c) datasets obtained from generalized additive modelling (GAM). Response shapes for the remaining predictors are illustrated in Figure S2. Predictors (see Sect. 2.2 for abbreviations) are presented in the descending order of their effect size in respective datasets. X-axis units appear in the original scale of the predictors. Y-axis displays partial residuals and labels the estimated degrees of freedom used in fitting the respective predictors to a response. Shaded areas depict 95 % confidence limits.**
