# Peer review of "New insights into the environmental factors controlling the ground thermal regime across the Northern Hemisphere: a comparison between permafrost and non-permafrost areas"

_The Cryosphere, 2018_

## Referee Comment (RC1) · Anonymous Referee #1 · 30 Jul 2018

Comments on "**New insights into the environmental drivers of the circumpolar ground thermal regime**" by Olli Karjalainen et al. submitted to The Cryosphere

**General**

This paper statistically related circumpolar observations of mean annual ground temperature (MAGT) and active-layer thickness (ALT) with climate, soil and vegetation variables. Based on the results, they provided some new insights into the major factors controlling the spatial distributions of MAGT and ALT. The analysis compiled a large number of circumpolar observations and the corresponding climate, soil and vegetation data, and the statistical modelling methods have not been seen often in permafrost studies. The results are interesting, especially by comparing the differences between permafrost and non-permafrost regions. I am not an expert of the statistical modelling methods. I assume they are valid and other reviewers can pay more attention to them.

**Major comments**

The analysis used thawing-degree-days (TDD) and freezing-degree-days (FDD) and other variables. It is valid for ALT since thawing occurs when air temperature (Tair) > 0 °C, and is related to TDD according to Stefan solution (especially in temporal variations). For AMGT, annual mean air temperature should be a major factor to consider. To assess the relative importance of cold season and warm season, winter mean and summer mean air temperatures are better choices than TDD and FDD since the length of the days is not a factor. I wonder why these factors were not chosen in the analysis. An important finding of this paper is that FDD were the main factor determining the spatial distribution of AMGT in the permafrost region while TDD dominated in the non-permafrost region. The days in a year when Tair < 0 °C are longer in permafrost region than in non-permafrost region. This difference automatically contributes to your results. If this effect is the major reason, I feel it is quite natural or understandable (the longer the more important) and should not be treated so sensationally as a significant finding. Any way, it would be meaningful and interesting to see the relationships between AMGT and annual mean, winter and summer mean Tair.

This study is to understand the factors affecting the spatial distributions of MAGT and ALT. The factors and mechanisms could be vey different from that controlling the temporal variations. The paper should make that clearer, including the title. It should be cautious about assuming factors controlling spatial distribution will automatically controlling the temporal changes (Lines 194-199). The paper used "divers", "driving" frequently. The words usually have a sense for temporal changes in climate change studies. For spatial distribution, it is better to avoid it. The relationships and impact indicators are based on statistical analysis. They depend on data, methods, and factors selected for analysis. It should be cautious to use the word "drive", just say "a factor has a close relationship with … or has large impacts statistically on …", especially when no strong physical processes and mechanisms to support the results. I like the phrase "new insights" in the title. The text should keep that cautious sense in the text.

**Some minor points**

Line 9: "The thermal dynamics of permafrost shape Earth surface systems and human activity in the Arctic …". "The thermal dynamics of permafrost" means temporal changes, which is different from the focus of the paper (spatial distribution). The word "shape" probably overstated the importance of the thermal dynamics.

As mentioned above, the title: "…driver of the circumpolar ground thermal region", Line 16: "main driver of MAGT in permafrost conditions" and similar sentences other places. These sentences give me a sense that they are drivers of temporal changes rather than factors influencing or determining spatial distributions of MAGT and ALT. The paper should make that clearer.

Line 17-19: The last sentence of the abstract is about temporal changes the authors like to infer from spatial patterns to temporal changes. It is problematic as I mentioned above. The term "initial ground thermal conditions" is not very clear, probably should say "the current ground thermal conditions". "local-scale topography-soil-driven variability", probably should be "local-scale variability in soil and topography" or simply "local-scale soil and topography".

Line 21: "geocryological development", the word "development" probably should be "dynamics".

Line 28: "activity", using plural.

Line 36: "ground temperatures are higher than air", adding "temperature" after "air", or "ground is warmer than air".

Line 59: "geographically comprehensive datasets of field-quantified MAGT (n = 784) and ALT (n = 298) observations.". Feels strange. How about "circumpolar field observations of MAGT and ALT". The number of sites in brackets can be described in methods section.

Line 63: "possible variation …", not very clear/direct. Using "differences" instead of "variation".

Lines 71: "MAGT values shallower than two meters …" should be "MAGT measured at less than two meters …" . Delete "systematically".

Line 85: "presenting", should be "representing"

Lines 94-98: You calculated TDD and FDD based on monthly climate data. Did you interpolated to daily or directly based on monthly averages? It is generally ok directly using monthly data based on the test of Frauenfeld et al. (2007. doi: 10.1002/joc.1372). You may refer to this paper for proof.

The tables and the figures are quite interesting. However, I feel the result section is a bit weak. I hope it can provide more detailed description, explanation and analysis of the tables and Figures.

Line 184: This study did not include any sites with permanent snow cover, you may delete "or permanent snow cover".

In discussion section, temporal changes were inferred from the spatial statistical results. The author should be cautious about that and clearly indicate the assumption.

Figures 2a-c: one color legend probably is enough. Figure 2d: the units of TDD and FDD should be °C d.

Figure 4: I am not familiar with the GAM. It would be useful to briefly describe how the response shapes are calculated and what do they mean? What is the unit of solar radiation?

---

## Referee Comment (RC2) · Anonymous Referee #2 · 3 Aug 2018

General Comments

This paper uses ground temperature and active layer data acquired from various sources to determine their relationships with various climate and other parameters to gain insights into the environmental drivers of the circumpolar ground thermal regime. While the analysis of this rather large data set is interesting (although others e.g. Peng et al. 2018 have made use of similar data sets) some of the insights regarding the various relationships are not necessarily new and have been reported elsewhere. In addition, some of the conclusions would appear to be at odds with those of other studies, which may be partly an issue of scale. A number of comments are offered below. These concerns should be addressed before the manuscript is considered for publication.

Some of the relationships considered in this paper particularly those concerning air temperature and soil conditions have been well summarized in key equations such as the Stefan equation and the TTOP equation and their variants (see for eg. Brown et al. 2000; Harland and Nixon, 1978; Hinkel and Nelson, 2003; Nelson and Outcalt 1993; Romanovsky and Osterkamp, 1995; Risborough and Smith 1998; Smith and Riseborough 2002). There have also been a number of studies over the last decade, including those at local to continental scales, that have considered permafrost-climate relations (i.e. consideration of ground temperature and active layer thickness) and role of various local factors (e.g. Romanovsky et al. 2010, 2017; Smith et al. 2009, 2010, 2012; Burn and Kokelj 2009; Palmer et al. 2012; Throop et al. 2012; Morse et al. 2012 etc).

The broad scale of the analysis and lack of site specific data likely obscures some of the important relationships between MAGT and various local factors such as vegetation, snow cover and terrain conditions (including properties of the earth materials). Studies over 40 years ago showed the relevance of these factors and their influence on the ground thermal regime and also the occurrence of permafrost, i.e. whether MAGT is above or below 0°C (e.g. Brown 1965, 1973; Nicholson and Granberg 1973; Thie 1974). The importance of substrate conditions (thermal properties, moisture content) is described in the thermal offset component in the TTOP model. The thermal offset which, under equilibrium conditions, is due to a difference between frozen and unfrozen thermal conductivity, can result in subsurface temperatures being below 0°C, and therefor the existence of permafrost, even though the ground surface temperature is above 0°C (see Romanovsky and Osterkamp, 1995; Risborough and Smith 1998; Smith and Riseborough 2002). This effect along with latent heat effects can result in the persistence of permafrost under warm climate conditions (e.g. Romanovsky et al. 2010; James et al. 2013).

This paper largely considers spatial variation in ALT and MAGT rather than temporal variations and the authors should be careful in making conclusions regarding future

changes in these variables in response to a changing climate. Also, a number of papers (such as Romanovsky et al. 2010, 2017; Smith et al. 2010 and others mentioned above) have considered the temporal variation in the ground thermal regime in the permafrost region and the factors affecting the response to a changing climate. In particular, these other studies have made conclusions regarding the importance of the initial ground thermal regime (i.e. how close MAGT is to 0°C and the importance of latent heat effects), the effect of snow cover, vegetation and substrate or soil conditions.

A large part of the paper appears to focus on the permafrost regions. However, the MAGT data utilized extends well beyond the permafrost regions and the cryospheric aspect (such as the seasonal frost depth) is not really considered in these more southerly regions and might be negligible in some areas. Given this is a journal focussed on the cryosphere and there appears to be a significant focus in the MS on permafrost, it is not clear why these additional sites were included in the analysis.

Specific Comments

Line 25-26 – One could argue that it is the presence of ground ice that influences the geomorphological processes and the impact of changing permafrost conditions.

Line 29 – Snow cover or snow depth is as (or perhaps more) important as precipitation with respect to the ground thermal regime.

Line 37-38 – One could argue it is the moisture content and drainage that are the important factors.

Line 39-41 – Romanovsky et al. (2010) is probably a better reference to use here for the role of latent heat in determining the response of the ground thermal regime to changes in air temperature.

Line 46-47 – As mentioned above, there have been circumpolar and continental analyses of the environmental drivers.

Line 52-54 – This observation wasn't unknown prior to Peng et al (2018) and as mentioned above, these relationships and the relevance of the "edaphic factors" are describe in variants of the Stefan Equation (e.g Harlan and Nixon, 1978; Nelson et al. 2000). Also, in an investigation of air temperature – ALT relationships across a range of ecoclimate zones, Smith et al. (2009) showed that the relationship varied according to vegetation and soil conditions (i.e. the edaphic factors).

Line 67 – Was ALT only obtained through mechanical probing or were some values acquired through analysis of shallow ground temperature records. In the results section you give a maximum value of ALT of >7 m and it is unlikely that this was determined through mechanical probing. Some of the reports used for sources of ALT data may report ALT determined by methods other than probing (including thaw tubes and ground temperature measurements). Note also that probing does not necessarily capture the maximum thaw depth.

Line 68 – The depth of ZAA can be much greater than 15 m and will depend on thermal properties of the subsurface materials. ZAA depth can be greater than 20 m for example in bedrock (see for eg. Romanovsky et al. 2010; Smith et al. 2010; Throop et al. 2012).

Line 60-83 – It is unclear whether the analysis utilizes a mean value for the entire 2010-2014 period for ground temperature, ALT, air temperature etc.

Line 95-98 – Snow depth can be highly variable in northern environments depending on exposure to wind and vegetation. This is a site specific factor and its influence is probably not adequately considered by only utilizing precipitation records.

Line 104-106 – What is the resolution?

Line 151-152 – This relationship was not unknown and has been shown by others (a couple of examples Brown, 1967, Throop et al. 2012 and GSC Open File 3954 available through GEOSCAN).

Line 153-154 – As shown in Smith et al. (2009), there is a more direct relationship

between TDD and ALT for tundra sites compared to vegetated sites or organic terrain.

Line 161-169 – Aren't these factors inter-related?

Line 180-187 – I would disagree that this finding about the effects of TDD and FDD is all that significant. Cold conditions are a requirement for permafrost so FDDair will have a higher value in permafrost environments compared to non permafrost environments. This is described by the Frost Index model of Nelson and Outcalt (1983).

Line 181 – Do you really mean there is a negative energy balance or do you mean FDD>TDD which is not the same thing (a negative energy balance would mean there is cooling over time).

Line 188-190 – See earlier comment regarding relationship between TDD and ALT and its variability with vegetation etc.

Line 189-193 – High Arctic sites do not necessarily have decimeter thaw depths. Greater thaw depths can be found in bedrock so the material type is important. Also, if thaw depths are largely obtained by probing there may be some bias in the data set as the method is limited by soil type (difficult or impossible in granular material and bedrock) and the depth of probing. As mentioned in previous comments, site specific factors are an important influence on ALT and its relationship with TDD and this is likely masked in your analysis.

Line 200-209 – The results presented don't really allow attribution of the effect of precipitation to advection over latent heat. Drainage will be an important factor.

Line 210-217 – As mentioned above, the amount of snow on the ground (snow depth) is probably the more important factor and is highly variable. Other studies have utilized winter n-factors to account for this effect in investigations of climate-ground temperature relationships. (see for example Morse et al. 2012; Palmer et al. 2012; Throop et al. 2012 as well as those cited in the MS).

Line 218-225 – A general northward decrease in MAGT which is associated with decreasing solar radiation and air temperature has been reported elsewhere (e.g Brown 1967, Smith et al. 2010; Romanovsky et al. 2010). As others have pointed out (see various papers already cited) the relationship is modulated by local factors. The incoming solar radiation that reaches the ground surface, and therefore influences the ground surface temperature and the deeper thermal regime, is probably the more important variable and probably not well captured in your data set.

Line 223-224 – Other studies (see those cited earlier) conclude that vegetation and soil properties are an important influence on the response of MAGT to changes in climate and therefore predictions of future conditions. Also, soil properties are an important influence on the thermal offset (which is not mentioned in this MS) which can be an important factor determining whether permafrost exists or not under warmer conditions (See previous comments).

271-272 – This has been concluded in other studies as mentioned in earlier comments.

Line 273-275 – See earlier comments regarding importance of substrate conditions (soil or rock properties) in influencing the response of the ground thermal regime (and future permafrost conditions) to changes in climate.

References

Burn CR, Kokelj SV (2009) The environment and permafrost of the Mackenzie Delta area. Permafrost and Periglacial Processes 20 (2):83-105. doi:10.1002/ppp.655

Brown J, Hinkel KM, Nelson FE (2000) The Circumpolar Active Layer Monitoring (CALM) Program: research designs and initial results. Polar Geography 24 (3):165-258

Brown RJE (1965) Some observations on the influences of climatic and terrain features on permafrost at Norman Wells, N.W.T., Canada. Canadian Journal Earth Science 2:15-31

Brown RJE (1967) Permafrost in Canada, Map 1264. Geological Survey of Canada,

[Figure]

National Research Council of Canada, Ottawa

Brown RJE 1973. Influence of climatic and terrain factors on ground temperatures at three locations in the permafrost region of Canada. In: Proceedings of the Second International Conference on Permafrost. North American Contribution, National Academy of Sciences, Washington, D.C., pp 27-34

Nicholson FH, Granberg HB 1973. Permafrost and snow cover relationships near Schefferville. In: Proceedings of the Second International Conference on Permafrost, North American Contribution. National Academy of Sciences, Washington, D.C., pp 151-158

Harlan, RL and Nixon JF 1978. Ground thermal regime. In OB Andersland and DM Anderson, eds., Geotechnical Engineering for Cold Regions, pp 103-163.

Hinkel KM, Nelson FE (2003) Spatial and temporal patterns of active layer thickness at Circumpolar Active Layer Monitoring (CALM) sites in northern Alaska 1995-2000. Journal Geophysical Research 108 (D2):8168. doi:10.1029/2001JD00097

James M, Lewkowicz AG, Smith SL, Miceli CM (2013) Multi-decadal degradation and persistence of permafrost in the Alaska Highway corridor, northwest Canada. Environmental Research Letters 8 045013:10. doi:10.1088/1748-9326/8/4/045013

Nelson F, Outcalt SI 1983. A frost index number for spatial prediction of ground-frost zones. In: Proceedings of the Fourth International Permafrost Conference, pp 907-911

Morse PD, Burn CR, Kokelj SV (2012) Influence of snow on near-surface ground temperatures in upland and alluvial environments of the outer Mackenzie Delta, Northwest Territories. Canadian Journal Earth Sciences 49:895-913. doi:10.1139/E2012-012

Palmer MJ, Burn CR, Kokelj SV (2012) Factors influencing permafrost temperatures across tree line in the uplands east of the Mackenzie Delta, 2004–2010. Canadian Journal of Earth Sciences 49:877-894. doi:10.1139/E2012-002

Riseborough DW, Smith MW 1998. Exploring the limits of permafrost. In: Proceedings of Seventh International Conference on Permafrost, Yellowknife, Canada. Collection Nordicana pp 935-941

Romanovsky V, Isaksen K, Drozdov D, Anisimov O, Instanes A, Leibman M, McGuire AD, Shiklomanov N, Smith SL, Walker D (2017) Chapter 4, Changing permafrost and its impacts. In: Snow, Water, Ice and Permafrost in the Arctic (SWIPA) 2017. Arctic Monitoring and Assessment Program (AMAP) Oslo, Norway, pp 65-102

Romanovsky VE, Smith SL, Christiansen HH (2010) Permafrost thermal state in the polar Northern Hemisphere during the International Polar Year 2007-2009: a synthesis. Permafrost and Periglacial Processes 21:106-116

Romanovsky VE, Osterkamp TE (1995) Interannual variations of the thermal regime of the active layer and near-surface permafrost in northern Alaska. Permafrost and Periglacial Processes 6:313-315

Smith MW, Riseborough DW (2002) Climate and limits of permafrost: a zonal analysis. Permafrost and Periglacial Processes 13:1-15

Smith SL, Romanovsky VE, Lewkowicz AG, Burn CR, Allard M, Clow GD, Yoshikawa K, Throop J (2010) Thermal state of permafrost in North America - A contribution to the International Polar Year. Permafrost and Periglacial Processes 21:117-135. doi:10.1002/ppp.690

Smith SL, Throop J, Lewkowicz AG (2012) Recent changes in climate and permafrost temperatures at forested and polar desert sites in northern Canada. Canadian Journal of Earth Sciences 49:914-924. doi:10.1139/E2012-019

Smith SL, Wolfe SA, Riseborough DW, Nixon FM (2009) Active-layer characteristics and summer climatic indices, Mackenzie Valley, Northwest Territories, Canada. Permafrost and Periglacial Processes 20 (2):201-220. doi:10.1002/ppp.651

Thie J (1974) Distribution and thawing of permafrost in the southern part of the discontinuous permafrost zone in Manitoba. Arctic 27:189-200

Throop J, Lewkowicz AG, Smith SL (2012) Climate and ground temperature relations at sites across the continuous and discontinuous permafrost zones, northern Canada. Canadian Journal of Earth Sciences 49:865-876. doi:10.1139/E11-075

---

## Author Response (AR1)

Dear Editor,

We want to thank both the reviewers for their constructive and helpful comments on manuscript TC-2018-144: *"New insights into the environmental drivers of the circumpolar ground thermal regime"*. We have carefully addressed all the comments and performed corresponding changes to the manuscript. This author's response document includes detailed responses to both reviewers' comments followed by a 'track changes' manuscript.

More precisely, we clarified the aim of our study and justified its place and novelty values in the long line of relevant research. In a few places, we provided a more detailed explanation of the methodology and the data used in the modelling. In addition, we included new discussions and supporting analyses concerning data-related issues pointed out by the reviewers.

Overall, we see that the revision has significantly improved the manuscript. We hope that our revised manuscript fully considered all the concerns raised by the reviewers and could be considered for publication in *The Cryosphere*.

Sincerely,
Olli Karjalainen (on behalf of all authors)

**Authors' response to reviewers' comments on the manuscript TC-2018-144: "New insights into the environmental drivers of the circumpolar ground thermal regime"**

Referee comments appear in gray,
author responses in black,
and suggested revisions to the original text are *italized*.

To facilitate effortless review we created a notation, in which each comment by the both referees was coded, e.g., **R1C1** = Referee#1, Comment#1.

Line numbers refer to the included track changes manuscript.

Referee #1

Comments on "New insights into the environmental drivers of the circumpolar ground thermal regime" by Olli Karjalainen et al. submitted to The Cryosphere

**General**

**R1C1** This paper statistically related circumpolar observations of mean annual ground temperature (MAGT) and active-layer thickness (ALT) with climate, soil and vegetation variables. Based on the results, they provided some new insights into the major factors controlling the spatial distributions of MAGT and ALT. The analysis compiled a large number of circumpolar observations and the corresponding climate, soil and vegetation data, and the statistical modelling methods have not been seen often in permafrost studies. The results are interesting, especially by comparing the differences between permafrost and non-permafrost regions. I am not an expert of the statistical modelling methods. I assume they are valid and other reviewers can pay more attention to them.

**R:** We thank the reviewer for the positive views.

**Major comments**

**R1C2** The analysis used thawing-degree-days (TDD) and freezing-degree-days (FDD) and other variables. It is valid for ALT since thawing occurs when air temperature (Tair) > 0 °C, and is related to TDD according to Stefan solution (especially in temporal variations). For AMGT, annual mean air temperature should be a major factor to consider. To assess the relative importance of cold season and warm season, winter mean and summer mean air temperatures are better choices than TDD and FDD since the length of the days is not a factor. I wonder why these factors were not chosen in the analysis. An important finding of this paper is that FDD were the main factor determining the spatial distribution of AMGT in the permafrost region while TDD dominated in the non-permafrost region. The days in a year when Tair < 0 °C are longer in permafrost region than in non-permafrost region. This difference automatically contributes to your results. If this effect is the major reason, I feel it is quite natural or understandable (the longer the more important) and should not be treated so sensationally as a significant finding. Any way, it would be meaningful and interesting to see the relationships between AMGT and annual mean, winter and summer mean Tair.

**R:** We appreciate these comments and recognize the need to fully address the issue.

We agree that mean annual air temperature (MAAT) is highly relevant for MAGT. However, MAAT cannot alone explain the variations in MAGT attributed to seasonal differences in the response of ground to air temperatures (e.g. Zhang et al. 1997; Smith et al. 2009), which need to be accounted for as the reviewer suggested. We consider TDD and FDD as suitable both 1) for examining the seasonal effects (as discussed at

lines 37–39), and 2) covering year-round climate forcing better than summer and winter mean T, inevitably missing some of the variability important for long-term averages of MAGT and ALT. Smith et al. (2009), for example, showed that a considerable part of thawing occurred outside June to August period in Mackenzie Valley, Canada. Moreover, using the same climatic parameters with MAGT and ALT allowed for comparisons between their controlling factors, which was one of the contributions of this study.

We performed additional analyses to examine the contributions of mean annual air temperature (MAAT), summer (June to August, JJA) and winter temperatures (December to February, DJF) to MAGT≤ 0°C. We tested the performances of models employing JJA+DJF in place of TDD+FDD, and also with MAAT as the only temperature predictor. Results are very similar to the original with TDD+FDD [average $R^2$=0.91 for calibration dataset, (0.85) for evaluation]; both JJA+DJF and MAAT models explain a marginally smaller part of variation in MAGT (Table I). Average RMSEs in both cases are higher than in the original models [0.95 (1.25)].

**Table I. Adjusted coefficient of determination ($R^2$) and root mean square error (RMSE) between observed and predicted mean annual ground temperature (MAGT) in calibration and evaluation (in brackets) datasets averaged over 100 permutations. The results are provided for datasets employing average air temperatures for summer (June, July and August, JJA) and winter (December, January and February, DJF), and mean annual air temperature (MAAT) as predictors.**

| Method | R2 (JJA+DJF) | RMSE (JJA+DJF) | R2 (MAAT) | RMSE (MAAT) |
|---|---|---|---|---|
| GLM | 0.84 (0.82) | 1.34 (1.42) | 0.85 (0.81) | 1.33 (1.42) |
| GAM | 0.86 (0.83) | 1.26 (1.38) | 0.86 (0.84) | 1.25 (1.34) |
| GBM | 0.93 (0.86) | 0.92 (1.24) | 0.92 (0.86) | 0.92 (1.24) |
| RF | 0.98 (0.87) | 0.53 (1.19) | 0.97 (0.87) | 0.55 (1.21) |
| Average | 0.90 (0.85) | 1.01 (1.31) | 0.90 (0.85) | 1.01 (1.30) |

TDD and JJA have almost perfect correlation, as do FDD and DJF (Figure I). Therefore, it is suggested that DD's well represent summer and winter conditions while also accounting for the climatic variability of the remaining year. In MAGT≤ 0°C dataset, MAAT had notably stronger correlation with DJF (0.86) than with JJA (0.40). This suggests that winter conditions contribute strongly to climatic forcing in permafrost regions even when length of the periods is not a factor. In non-permafrost conditions, MAAT had much more similar correlations with JJA (0.89) and DJF (0.94).

[Figure]

**Figure I. Spearman correlations for MAGT in permafrost (a) and non-permafrost conditions (b).**

Looking at effect sizes (Table II) and variable importance values (Table III) computed for DFJ and JJA, however, it is evident that summer temperatures here have a larger contribution than winter. Should this be due to unaccounted variability outside these months or the buffering effect of snow cover during DJF, we conclude that both TDD+FDD and JJA+DJF can be used to model the ground thermal regime with similar performance (TDD+FDD was slightly better in the light of model performance). However, we consider it is important to take into account the full 12-month climatic variability especially when assessing long-term averages of ground thermal regime at circumpolar scale, and thus prefer TDD+FDD.

**Table II. The effect size of individual predictors and their four-model averages**

| | MAGT≤ 0°C (°C) | | | | |
| --- | --- | --- | --- | --- | --- |
| | GLM | GAM | GBM | RF | Avg |
| **DJF** | 7.8 | 2.1 | 2.3 | 2.2 | 3.6 |
| **JJA** | 10.0 | 10.4 | 2.7 | 3.2 | 6.6 |
| **PrecipWater** | 2.9 | 3.1 | 5.2 | 3.5 | 3.7 |
| **PrecipSnow** | 4.9 | 4.6 | 0.1 | 0.3 | 2.5 |
| **SolarRad** | 3.6 | 4.0 | 0.0 | 0.3 | 2.0 |
| **CoarseSed** | 1.1 | 2.0 | 0.1 | 0.1 | 0.8 |
| **FineSed** | 0.8 | 0.6 | 0.1 | 0.4 | 0.5 |
| **SOC** | 0.4 | 0.6 | 0.6 | 1.0 | 0.6 |
| **NDVI** | 0.6 | 0.3 | 0.1 | 1.0 | 0.5 |

**Table III. Variable importance values for individual predictors.**

| MAGT pf | #FineSed | #CoarseSed | #NDVI | #SOC | #SolarRad | #PrecipWater | #PrecipSnow | #djf | #jja |
| --- | --- | --- | --- | --- | --- | --- | --- | --- | --- |
| GLM | 0.00 | 0.01 | 0.00 | 0.00 | 0.06 | 0.01 | 0.09 | 0.18 | 0.50 |
| GAM | 0.00 | 0.02 | 0.00 | 0.00 | 0.04 | 0.04 | 0.08 | 0.23 | 0.42 |
| GBM | 0.00 | 0.00 | 0.00 | 0.00 | 0.02 | 0.41 | 0.00 | 0.17 | 0.05 |
| RF | 0.00 | 0.00 | 0.01 | 0.00 | 0.04 | 0.17 | 0.01 | 0.09 | 0.06 |

**R1C3** This study is to understand the factors affecting the spatial distributions of MAGT and ALT. The factors and mechanisms could be vey different from that controlling the temporal variations. The paper should make that clearer, including the title. It should be cautious about assuming factors controlling spatial distribution will automatically controlling the temporal changes (Lines 194-199). Thepaper used "divers", "driving" frequently. The words usually have a sense for temporal changes in climate change studies. For spatial distribution, it is better to avoid it. The relationships and impact indicators are based on statistical analysis. They depend on data, methods, and factors selected for analysis. It should be cautious to use the word "drive", just say "a factor has a close relationship with … or has large impacts statistically on …", especially when no strong physical processes and mechanisms to support the results.

**R:** Our focus indeed is on spatial variation of MAGT and ALT rather than temporal dynamics. Therefore, we agree on avoiding "drivers", and preferring "factors" and other ways to express the relationships between the predictors and responses. We revised the text in any places where temporal changes were discussed (mainly the Abstract, Discussion and Conclusions), stressing the spatial focus of this study and the need to remain cautious when discussing temporal dynamics. However, we decided to leave some cautious discussions about the potential effects of warming climate on MAGT and ALT (lines 225–230, 234–237).

To follow the account we modified the title accordingly: *"New insights into the environmental factors controlling the circumpolar ground thermal regime"*.

**R1C4** I like the phrase "new insights" in the title. The text should keep that cautious sense in the text.
**R:** We thank the reviewer for this view and strive at remaining insightful.

**Some minor points**
**R1C5** Line 9: "The thermal dynamics of permafrost shape Earth surface systems and human activity in the Arctic …". "The thermal dynamics of permafrost" means temporal changes, which is different from the focus of the paper (spatial distribution). The word "shape" probably overstated the importance of the thermal dynamics.
**R:** As agreed upon (R1C3), we focus on examining the spatial variation of the ground thermal regime and revised accordingly (line 9–10): *"The thermal state of permafrost affects Earth surface systems and human activity in the Arctic and has…"*

**R1C6** As mentioned above, the title: "…driver of the circumpolar ground thermal region", Line 16:"main driver of MAGT in permafrost conditions" and similar sentences other places. These sentences give me a sense that they are drivers of temporal changes rather than factors influencing or determining spatial distributions of MAGT and ALT. The paper should make that clearer.
**R:** Great thanks for pointing out these occurrences. We revised these and related sentences throughout the manuscript to clarify the study's focus. Please see R1C3.

**R1C7** Line 17-19: The last sentence of the abstract is about temporal changes the authors like to infer from spatial patterns to temporal changes. It is problematic as I mentioned above. The term "initial ground thermal conditions" is not very clear, probably should say "the current ground thermal conditions". "local-scale topography-soil-driven variability", probably should be "local-scale variability in soil and topography" or simply "local-scale soil and topography".
**R:** These are highly valuable suggestions. The last sentence was revised accordingly (lines 19–22): *"Our findings suggest that in addition to climatic factors, local-scale variability in soil and topography need to be considered in order to realistically assess the current and future ground thermal regimes across the circumpolar region."*

**R1C8** Line 21: "geocryological development", the word "development" probably should be "dynamics".
**R:** We agree that "dynamics" is more unambiguous here. Replacement made.

**R1C9** Line 28: "activity", using plural.
**R:** Replaced "activity" → "activities"

**R1C10** Line 36: "ground temperatures are higher than air", adding "temperature" after "air", or "ground is warmer than air".
**R:** Revised accordingly (line 40): *"ground is warmer than air"*

**R1C11** Line 59: "geographically comprehensive datasets of field-quantified MAGT (n = 784) and ALT (n = 298) observations.". Feels strange. How about "circumpolar field observations of MAGT and ALT". The number of sites in brackets can be described in methods section.

**R:** The suggested simplification reads well and was revised with accordingly (line 66): *"...circumpolar field observations of MAGT and ALT."* The numbers of ALT sites were added in the Methods, line 77: *"and that ALT (n = 298) ..."*

**R1C12** Line 63: "possible variation …", not very clear/direct. Using "differences" instead of "variation".

**R:** We agree that "differences" is better here and have made a replacement (line 72): *"possible differences…"*

**R1C13** Lines 71: "MAGT values shallower than two meters …" should be "MAGT measured at less than two meters …" . Delete "systematically".

**R:** Very approvable suggestions, the revised sentence (lines 84–85) reads: *"MAGT measured at less than two meters below the surface were excluded unless reported to be at the depth of ZAA."*

**R1C14** Line 85: "presenting", should be "representing"

**R:** Prefix added.

**R1C15** Lines 94-98: You calculated TDD and FDD based on monthly climate data. Did you interpolated to daily or directly based on monthly averages? It is generally ok directly using monthly data based on the test of Frauenfeld et al. (2007. doi: 10.1002/joc.1372). You may refer to this paper for proof.

**R:** We calculated the indices based on monthly temperature averages utilising the WorldClim data. It indeed was our intention to cite Frauenfeld et al. to show that using monthly data provides very similar values to those derived from daily observations. We considered it necessary to elaborate the original manuscript (lines 109–110): *"Frauenfeld et al. (2007) showed that their use instead of daily temperatures accounts for less than 5 % error for most high-latitude land areas".*

**R1C16** The tables and the figures are quite interesting. However, I feel the result section is a bit weak. I hope it can provide more detailed description, explanation and analysis of the tables and Figures.

**R:** We appreciate the reviewer's views here. To give a more detailed account, we added some further explanations. However, we preferred to focus on main findings to not unnecessarily lengthen the manuscript or repeat the information conveyed by the figures and tables.

(lines 170–171): *"Coarse sediments and SOC, especially, were important and showed clear, yet non-linear, responses to ALT, respectively (Fig. 4c)."*

(177–181): *"On average, RMSEs were low (~1 °C) in MAGT≤0 °C and MAGT>0 °C calibration datasets. When predicted over evaluation datasets, the average increased slightly more in non-permafrost conditions. A similar increase of 40 % was documented with ALT. For each response, GBM and RF had lower RMSEs (i.e. higher predictive performance) than GLM and GAM, but also larger change between calibration and evaluation datasets, indicating that GLM and GAM produced more robust predictions."*

(195–198): *"Considering the remaining predictors, clear differences were observed in cases of SOC and NDVI, both higher in MAGT$_{>0\ °C}$ dataset. [...] In contrast to variable importance results (Fig. 3c), snow precipitation had a larger average effect than coarse sediments and SOC, both of which nevertheless had a considerable effect."*

**R1C17** Line 184: This study did not include any sites with permanent snow cover, you may delete "or permanent snow cover".

**R:** This is a spot-on notion. Deleted *"or permanent"* at line 210.

**R1C18** In discussion section, temporal changes were inferred from the spatial statistical results. The author should be cautious about that and clearly indicate the assumption.

**R:** Please see R1C3. We clarified the original text with a follow-up sentence (underlined, lines 223–224): *"Moreover, large inconsistencies between observed ALT and climate-warming trends have been documented (e.g. Wu et al., 2012; Gangodagamage et al., 2014). Although temporal dynamics of ALT are beyond our analyses, this suggests that thaw depth and air temperatures are, to a degree, decoupled by local conditions."*

**R1C19** Figures 2a-c: one color legend probably is enough. Figure 2d: the units of TDD and FDD should be °C d.

**R:** Two-color legend was used to show whether the significant correlations are positive (red) or negative (blue). This was to allow more easy visual interpretation of the interrelationships within and between a, b and c panels without having to compare the numbers. Therefore, we preferred not to remove the colors, but changed the unit of TDD/FDD to °C d here and in Supplementary Figures 1 and 2.

**R1C20** Figure 4: I am not familiar with the GAM. It would be useful to briefly describe how the response shapes are calculated and what do they mean?

**R:** This an important remark and we agree that describing response curve derivation will improve legibility. We described the calculation and interpretation of GAM response curves in the Methods (lines 134–135) *"The curves show smoothed fit between response and a predictor while all other predictors are fixed at their average (Hjort and Luoto, 2011)."*

**R1C21** What is the unit of solar radiation?

**R:** The unit of solar radiation was added at line 118: *"to estimate the potential incident solar radiation (PISR, W cm$^{-1}$ a$^{-1}$)…"*
**R2C1** This paper uses ground temperature and active layer data acquired from various sources to determine their relationships with various climate and other parameters to gain insights into the environmental drivers of the circumpolar ground thermal regime. While the analysis of this rather large data set is interesting (although others e.g. Peng et al. 2018 have made use of similar data sets) some of the insights regarding the various relationships are not necessarily new and have been reported elsewhere. In addition, some of the conclusions would appear to be at odds with those of other studies, which may be partly an issue of scale. A number of comments are offered below. These concerns should be addressed before the manuscript is considered for publication.

**R:** We thank the reviewer for highly expertized comments across the manuscript. Peng et al. (2018), along with Park et al. (2013, 2015), Luo et al. (2016), Guo & Wang (2017) and Zhang et al. (2018), have indeed used similar data sets. In addition to the unprecedented geographical coverage of MAGT sites, our dataset allowed for assessing the differences between permafrost and non-permafrost regions, to our knowledge, previously not (semi)quantitatively studied at this extent. We consider that the novelty of our work lies in its implementation of multiple statistical modelling techniques from regression to decision-tree-based machine learning, and the high spatial resolution (~1 km$^2$ grid-cell size) combined with circumpolar extent the approach offers.

Our view to ground thermal regime modelling implies that we break the system into components (predictors; climatic and local factors) and assess their correlative relationships with MAGT and ALT. Some advantages of statistical models are that they are more cost-efficient than mechanistic models (which currently have limited high-resolution applicability in hemisphere scale investigations), and enable examining permafrost-climate relations without pre-defined parameters, e.g., thermal diffusivity. Unlike in mechanistic transient models (which can arguably provide more accurate regional predictions than statistical methods) we do not model processes, such as phase transition in the Stefan equation, but focus on the individual predictors' effects. In addition, it can account for variables related to topography and land cover (vegetation) that could be difficult to otherwise parametrize.

Importantly, our analyses go beyond examining the relative contributions of the predictors. The effect size analysis provides numerical information about the effective magnitudes of the relationships. In addition, the response curves provide means to assess the shape of the response (direction and non-linearity) facilitating the understanding of the observed responses. To better point out the novelties of this study, we reflected these aspects in the Abstract (lines 11–15, edits underlined): *"Here, we statistically related circumpolar observations of mean annual ground temperature (MAGT) and active-layer thickness (ALT) to high-resolution (~1 km2) geospatial data of climatic and local environmental conditions. The aim was to characterize the relative importance of key environmental factors and the magnitude and direction of their effects in predicting the circumpolar ground thermal regime at 1-km scale."*

Introduction (lines 62–68): *"More specifically, we aim to (1) calibrate realistic models of MAGT and ALT (the responses) utilizing geospatial data on climatic and local conditions (the predictors) across the Northern Hemisphere land areas, and (2) examine the nature of the contributing factors in both permafrost and non-permafrost conditions using circumpolar field observations of MAGT and ALT. The analyses provide detailed insights into the importance of key environmental factors and the magnitudes and direction of their effect at 1-km resolution."*

and in the Conclusions (lines 313–315): *"In permafrost conditions, different key factors accounted for variation in MAGT and ALT; climate was paramount for MAGT, while local environmental conditions were emphasized in case of ALT."* and lines 320–323: *"In addition to providing theoretical insights about effective magnitudes and directions of the key contributing factors at circumpolar scale, multi-variate modelling frameworks capable of employing high-resolution geospatial data are valuable for the spatio-temporal prediction of ground thermal regime at the circumpolar scale. ..."*

The scale, as the reviewer suggests, presumably is behind some discrepancies with the previous studies, which is one reason why we think it is interesting and important to do research from this viewpoint. Climate and ground temperature relations differ between sites in different regions as shown by e.g. Throop et al. (2012), and some of these may average-out, but our aim was to quantify how the relevant factors affect at circumpolar scale, the contribution of which provides new information for future research.

**R2C2** Some of the relationships considered in this paper particularly those concerning air temperature and soil conditions have been well summarized in key equations such as the Stefan equation and the TTOP equation and their variants (see for eg. Brown et al. 2000; Harland and Nixon, 1978; Hinkel and Nelson, 2003; Nelson and Outcalt 1993; Romanovsky and Osterkamp, 1995; Risborough and Smith 1998; Smith and Riseborough 2002). There have also been a number of studies over the last decade, including those at local to continental scales, that have considered permafrost-climate relations (i.e. consideration of ground temperature and active layer thickness) and role of various local factors (e.g. Romanovsky et al. 2010, 2017; Smith et al. 2009, 2010, 2012; Burn and Kokelj 2009; Palmer et al. 2012; Throop et al. 2012; Morse et al. 2012 etc).

**R:** We appreciate the reviewer's efforts in providing an extensive listing of relevant references. We have acquainted with most of this literature and used it as a basis for our modelling framework. We acknowledge the findings of the large-scale studies cited by the reviewer here (Romanovsky et al. 2010, Smith et al. 2010; Throop et al. 2012) as well as those in R2C1 and its response, but argue that our analyses provide new detailed information (relative importance, effect size, shape of response) about the contributing factors. Please see R2C1.

**R2C3** The broad scale of the analysis and lack of site specific data likely obscures some of the important relationships between MAGT and various local factors such as vegetation, snow cover and terrain conditions (including properties of the earth materials). Studies over 40 years ago showed the relevance of these factors and their influence on the ground thermal regime and also the occurrence of permafrost, i.e. whether MAGT is above or below 0_C (e.g. Brown 1965, 1973; Nicholson and Granberg 1973; Thie 1974). The importance of substrate conditions (thermal properties, moisture content) is described in the thermal offset component in the TTOP model. The thermal offset which, under equilibrium conditions, is due to a difference between frozen and unfrozen thermal conductivity, can result in subsurface temperatures being below 0_C, and therefor the existence of permafrost, even though the ground surface temperature is above 0_C (see Romanovsky and Osterkamp, 1995; Risborough and Smith 1998; Smith and Riseborough 2002). This effect along with latent heat effects can result in the persistence of permafrost under warm climate conditions (e.g. Romanovsky et al. 2010; James et al. 2013).

**R:** This is an enjoyable summary of decades' worth of research. For discussions about the potential effect of scale, please see (R2C1). The lack of site-specific data is an obvious source of uncertainty, and it would be indeed interesting to see how geospatial data compares to measured values at the used sites should they be sufficiently available. However, we would like to point out that vegetation, snow cover and terrain conditions indeed had an effect on the ground thermal regime in permafrost conditions albeit it was clearly smaller than that of air temperature. As pointed out above (see R2C1 for performed edits) our results do not just echo the relevance of, for example air temperature, but also show the magnitude and shape of the relationship between ground thermal regime and environmental factors. Thus, we consider that our results have substantial added value to the study of the permafrost-environment relationships.

We decided to diverge from the TTOP model and its offset components by independently addressing the determinants of these offsets (soil properties, snow). Therefore, also in our analyses, joint effect of all considered factors is suggested to be able to predict subzero MAGT even when surface temperature is above zero, as happens when thermal offset is strong. As a sign of this, the latent-heat effect can be discerned in the response shapes (Fig. 4a) as a flattened curve near 0 °C, that is, weakened relationship between FDD and MAGT.

**R2C4** This paper largely considers spatial variation in ALT and MAGT rather than temporal variations and the authors should be careful in making conclusions regarding future changes in these variables in response to a changing climate. Also, a number of papers (such as Romanovsky et al. 2010, 2017; Smith et al. 2010 and others mentioned above) have considered the temporal variation in the ground thermal regime in the permafrost region and the factors affecting the response to a changing climate. In particular, these other studies have made conclusions regarding the importance of the initial ground thermal regime (i.e. how close MAGT is to 0_C and the importance of latent heat effects), the effect of snow cover, vegetation and substrate or soil conditions.
**R:** This is a highly valuable argument, and was pointed out by the reviewer #1 (R1C3) as well. We recognize the need to more clearly state that our focus is in spatial variation in ALT and MAGT and not on temporal dynamics of permafrost. See R2C1 for discussions about the added information from our study compared to the previous. We carefully revised each sentence where temporal effects were assessed based on our results (lines 220–221, 315–318; R1C5, R1C7, R1C18).

**R2C5** A large part of the paper appears to focus on the permafrost regions. However, the MAGT data utilized extends well beyond the permafrost regions and the cryospheric aspect (such as the seasonal frost depth) is not really considered in these more southerly regions and might be negligible in some areas. Given this is a journal focussed on the cryosphere and there appears to be a significant focus in the MS on permafrost, it is not clear why these additional sites were included in the analysis.
**R:** Seasonal frost depth would be another important issue to address at broad scale using e.g. statistical modelling framework. Here, we focus on MAGT because it can provide comparable information on ground thermal conditions inside and outside the permafrost regions. Although our focus is on permafrost, we performed similar analyses for non-permafrost regions to test the hypothesized influence of current ground thermal conditions (presence or absence of permafrost) on the effect of controlling factors. This information is important in the context of changing future permafrost extent; if currently frozen areas thaw, their response to environmental forcing is altered.

**Specific Comments**
**R2C6** Line 25-26 – One could argue that it is the presence of ground ice that influences the geomorphological processes and the impact of changing permafrost conditions.
**R:** At this early stage of the Introduction, we wanted to first point out broader factors and not specific factors such as ground ice, soil properties or topography. However, ground ice is central for geomorphological processes, and thus we added a mention on this (lines 46–47): *"In addition to the effect of ground ice content on heat transfer, its development is an important geomorphic factor (e.g. Liljedahl et al., 2016)."*

**R2C7** Line 29 – Snow cover or snow depth is as (or perhaps more) important as precipitation with respect to the ground thermal regime.
**R:** Here, we considered snowfall to be included in "precipitation" related to our method of deriving snowfall variable (PrecipSnow) from precipitation data. To reduce ambiguity, we decided to only refer to climatic conditions here (line 33): *"Climatic conditions account for large-scale…"*. Also at line 36, we moved snow

before soil and vegetation to stress its importance: *"… intercepting layers of snow, soil and vegetation mediate their effect…"*

**R2C8** Line 37-38 – One could argue it is the moisture content and drainage that are the important factors.
**R:** This is very true, but in the Introduction we would prefer characterizing the studied factors and their major effects. However, we added a sentence to elaborate how fine-scale factors affect soil moisture (and snow) distribution in the Discussions (lines 298–300): *"Fine-scale biophysical factors affecting drainage conditions and distribution of wind-drifted snow (e.g. vegetation and small topo-graphic depressions) are largely averaged-out and cannot be accounted for at 1-km resolution."*

**R2C9** Line 39-41 – Romanovsky et al. (2010) is probably a better reference to use here for the role of latent heat in determining the response of the ground thermal regime to changes in air temperature.
**R:** We replaced Ekici et al. (2015) with Romanovsky et al. (2010) (line 45).

**R2C10** Line 46-47 – As mentioned above, there have been circumpolar and continental analyses of the environmental drivers.
**R:** Although our approach offers new ways to examine these environmental factors (variable importance, effect size, shape of response; please see R2C1) with very comprehensive observational data over the Northern Hemisphere north of 30[th] latitude, we acknowledge that the present study and relevant previous efforts are hard to compare related to varying extent, spatial resolution, used methods, observational data etc. Therefore, statements concerning lack of relevant studies can be ambiguous, and we thus remove the sentence (lines 51–53). We believe that the novelty of our study becomes obvious elsewhere.

**R2C11** Line 52-54 – This observation wasn't unknown prior to Peng et al (2018) and as mentioned above, these relationships and the relevance of the "edaphic factors" are describe in variants of the Stefan Equation (e.g Harlan and Nixon, 1978; Nelson et al. 2000). Also, in an investigation of air temperature – ALT relationships across a range of ecoclimate zones, Smith et al. (2009) showed that the relationship varied according to vegetation and soil conditions (i.e. the edaphic factors).
**R:** Our intention was not to claim that Peng et al were the first to stress this. Instead, we wanted to highlight the research need with recent pan-Arctic study and the recommendations given therein. We slightly modified the sentence (lines 58–60): *"Recently, Peng et al. (2018) assessed spatio-temporal long-term trends in circumpolar ALT with a large observational dataset stressing that ALT strongly depends on local topo-edaphic factors (e.g. Harlan and Nixon, 1978) and that thorough analyses of environmental factors controlling ALT at varying scales are urgently required."* A reference to Harlan and Nixon (1978, line 59) was added to acknowledge the long history of this observation.

Smith et al. study is very interesting in its setting, and would be interesting to reproduce with our modelling methods. We believe that here lies one reason for the discrepancies between our results and Smith et al. (and some others); at circumpolar scale, some region-specific controls can be averaged out. Therefore, we argue that moderately weak circumpolar ALT/TDD connection is valid, given the highly heterogeneous environmental conditions across the observed ALT sites. Nevertheless, our results show the multivariate nature of ALT and strongly point out the effects of soil conditions. Vegetation's effect, in turn, was vague possibly owing to the inadequacy of NDVI in depicting complex (seasonal) involved processes (see Anisimov & Sherstiukov 2016) as discussed in the original manuscript (lines 272–276).

**R2C12** Line 67 – Was ALT only obtained through mechanical probing or were some values acquired through analysis of shallow ground temperature records. In the results section you give a maximum value of ALT of >7 m and it is unlikely that this was determined through mechanical probing. Some of the reports used for sources of ALT data may report ALT determined by methods other than probing (including thaw tubes and ground temperature measurements). Note also that probing does not necessarily capture the maximum thaw depth.

**R:** We agree that the expression in the original manuscript was incomplete and see the need to elaborate the text (lines 77–79): *"...ALT (n = 298) values represented the maximum thaw depth of a given year based on mechanical probing or derived from ground temperature measurements or thaw tubes (Brown et al., 2000; Aalto et al., 2018a)."* The deepest values were measured from borehole temperatures.

**R2C13** Line 68 – The depth of ZAA can be much greater than 15 m and will depend on thermal properties of the subsurface materials. ZAA depth can be greater than 20 m for example in bedrock (see for eg. Romanovsky et al. 2010; Smith et al. 2010; Throop et al. 2012).

**R:** This is a valid point and should be disclosed in the text. We chose 15 m as a compromise value (to be systematic, only one depth was chosen), because it is usually included in the depth ranges for ZAA in general (e.g. French 2007). We elaborated the issue accordingly (underlined, lines 80–82): *"...at the depth of 15 m, where annual temperature fluctuation in most conditions is negligible (see French, 2007), although in thermally highly diffusive subsurface materials, such as bedrock, the depth can be greater (Throop et al. 2012)."* We added a reference to Throop et al. because it was an important reference work when writing the original manuscript but for some reason was not included in the submitted manuscript.

**R2C14** Line 60-83 – It is unclear whether the analysis utilizes a mean value for the entire 2010- 2014 period for ground temperature, ALT, air temperature etc.

**R:** We acknowledged the need to clarify the issue. MAGT and ALT values used in modelling were calculated from averages of all available full years of observations (or appropriate single measurements, i.e. at or near ZAA). This is now detailed in the text (lines 73–74): *"For each MAGT and ALT site, averages over the study period were then calculated from available annual averages or suitable single measurements."*

The WorldClim data were adjusted to represent 15-year monthly climate averages from the same period. We clarified this (lines 104–105): *"Monthly averages over this 15-year period were then used to derive the following climate parameters."*

NDVI was calculated from June to August imagery from 2000 to 2014, as stated at lines 113–115.

**R2C15** Line 95-98 – Snow depth can be highly variable in northern environments depending on exposure to wind and vegetation. This is a site specific factor and its influence is probably not adequately considered by only utilizing precipitation records.

**R:** We strongly agree on this, and consider that the lack of data on specific snow thickness or snow water equivalency is a limitation in our approach. However, suitable data on snow depth or water equivalent at this extent (i.e. the whole pan-Arctic area) and resolution (ca. 1 km) unfortunately are not available. Moreover, at this resolution fine-scale biophysical factors affecting wind-drifted snow are largely smoothed out as discussed earlier (R2C8).

**R2C16** Line 104-106 – What is the resolution?

**R:** All the predictor variables have a 30 arc-second resolution (~1 km$^2$) as stated at lines 98-100 (edits underlined): *"Nine geospatial predictors representing climatic (air temperature and precipitation) and local*

*(potential incident solar radiation, vegetation and soil properties) conditions at 30 arc-second spatial resolution were selected to examine their potential effects on MAGT and ALT at the circumpolar scale."*

**R2C17** Line 151-152 – This relationship was not unknown and has been shown by others (a couple of examples Brown, 1967, Throop et al. 2012 and GSC Open File 3954 available through GEOSCAN).

**R:** Great thanks for pointing out the references. The air temperature-permafrost relationship was reported here because it is central to the study and for the subsequent discussions, even though it is not a novel finding. Thus, we see no need to revise the text here.

**R2C18** Line 153-154 – As shown in Smith et al. (2009), there is a more direct relationship between TDD and ALT for tundra sites compared to vegetated sites or organic terrain.

**R:** Although the ALT-TDD relationship was not very strong at circumpolar scale, the moderately high contribution of soil organic content on ALT represents the strong thermal offset at sites where organic layer is thick (Table 2, Fig. 4). This, along with addressing other factors affecting the TDD-ALT relationship, were discussed in the Discussion section (lines 262–271). The study by Smith et al. was also cited therein, and thus no changes were seen necessary.

**R2C19** Line 161-169 – Aren't these factors inter-related?

**R:** Effect size analysis is capable of addressing the individual predictor contributions by recursively averaging the values of the others in the computation (described in Section 2.3.4.). Thereby, each predictors' individual effect can be assessed. No changes were considered necessary.

**R2C20** Line 180-187 – I would disagree that this finding about the effects of TDD and FDD is all that significant. Cold conditions are a requirement for permafrost so FDDair will have a higher value in permafrost environments compared to non permafrost environments. This is described by the Frost Index model of Nelson and Outcalt (1983).

**R:** We agree that it was anticipated that freezing temperatures would dominate in a colder area. We think it is important to report the effectiveness of FDD in permafrost regions, and TDD elsewhere, because these findings confirm previous understanding of climate-permafrost relationship at circumpolar setting. However, based on our results we can also characterize these relationships (please see discussions in R2C1) and thereby improve previous understanding rather than only confirming it.

Consequently we revised the text at lines 204–206: *"Our results show are in line with previous understanding that climatic conditions are the primary factors affecting the long-term averages of circumpolar MAGT at 1-km resolution but also indicate that the effects of TDD and FDD on MAGT are dependent on the current permafrost occurrence."* and added a notion (underlined) on the directness of the response between MAGT<0°C and TDD (line 208–209): *"At sites without permafrost, TDD has the dominant nearly linear (Fig. 4b) effect…"*

**R2C21** Line 181 – Do you really mean there is a negative energy balance or do you mean FDD>TDD which is not the same thing (a negative energy balance would mean there is cooling over time).

**R:** Great thanks for your comment. We have improved the text, we completely revised this sentence. It is true that we cannot make statements about the current state of energy balance at the sites. Here, with negative energy balance we referred to the initial process behind the conditions that permafrost exists and did not mean to argue that FDD>TDD means negative energy balance. Revised at lines 206–208: *"As anticipated, FDD has*

*higher influence on MAGT in permafrost conditions where strong freezing is a prerequisite for the occurrence of permafrost (e.g. Smith & Riseborough, 1996)."*

**R2C22** Line 188-190 – See earlier comment regarding relationship between TDD and ALT and its variability with vegetation etc.
**R:** Please see R2C18.

**R2C23** Line 189-193 – High Arctic sites do not necessarily have decimeter thaw depths. Greater thaw depths can be found in bedrock so the material type is important. Also, if thaw depths are largely obtained by probing there may be some bias in the data set as the method is limited by soil type (difficult or impossible in granular material and bedrock) and the depth of probing.
**R:** We recognized the need to clarify the text. By high-Arctic sites we referred to low-lying regions in the circumpolar north. High latitudes obviously also have mountains with highly conductive ground and therefore thick ALT. The revised sentence reads (lines 215–222): "*According to our results, the spatial linkage is more elusive at a broader scale and could be attributed to the great circumpolar variation in ALT. The majority of high-Arctic sites locate on low-lying tundra overlaid by mineral and organic soil layers, whereas at mid-latitudes (the Alps, central Asian mountain ranges) permafrost predominantly occurs in mountains with thin soils and thermally diffusive bedrock. This difference partly explains generally small and large ALT within the respective regions notwithstanding that they can have similar average climatic conditions (e.g. TDD, see Fig. 2d).*"

As discussed (R2C12), ALT measurements have been performed by probing or derived from boreholes and thaw tubes. We would like to point out that compiled data came from established sources often cited in permafrost science. We nevertheless acknowledged the potential source of uncertainty in probing in coarse material. Therefore, we excluded all the probing measurements that reportedly did not reach the maximum depth of thaw as reported by sources. To our opinion there is no need revise the text here.

**R2C24** As mentioned in previous comments, site specific factors are an important influence on ALT and its relationship with TDD and this is likely masked in your analysis.
**R:** Our aim was to examine the individual influences of climate and local factors on MAGT and ALT. The results show that soil factors indeed had notable individual contributions on ALT. TDD-ALT relationship therefore is not necessarily masked, but rather stress that at this scale TDD is not the major control of spatial variability. The weak connection between observed ALT and TDD is also visible in Figure 2d. Please see R2C18.

Based on the comment, we discussed this issue in more detail (lines 216–219) in R2C23.

**R2C25** Line 200-209 – The results presented don't really allow attribution of the effect of precipitation to advection over latent heat. Drainage will be an important factor.
**R:** This is an important point; we need to be cautious when assessing any subsurface processes as they were not directly studied. Here we relate our findings to previously documented connections including the effects discussed in the original manuscript. The role of precipitation and related mechanisms at this scale needs more studying (cf. Peng et al. 2018).

We revised this paragraph (lines 230–242) by moving the discussions about advective heat to the end of the paragraph and by giving a stronger emphasis on latent heat effect reinforced by the amount of water precipitation. Westermann et al. (2011) was cited to accompany this (lines 234–237, underlined): *"Projected*

*greater proportion of liquid precipitation (e.g. AMAP, 2017; Bintanja and Andry, 2017) potentially has a direct effect on the ground thermal regime through its influence on latent heat exchange (Westermann et al., 2011), and convective warming during spring (Kane et al., 2001) and summertime (Melnikov et al., 2004; Marmy et al., 2013)."*

**R2C26** Line 210-217 – As mentioned above, the amount of snow on the ground (snow depth) is probably the more important factor and is highly variable. Other studies have utilized winter n-factors to account for this effect in investigations of climate-ground temperature relationships. (see for example Morse et al. 2012; Palmer et al. 2012; Throop et al. 2012 as well as those cited in the MS).

**R:** As discussed in the original manuscript, the relatively low contribution of snowfall contradicted with previous studies. Please see discussion concerning data limitations (R2C15).

**R2C27** Line 218-225 – A general northward decrease in MAGT which is associated with decreasing solar radiation and air temperature has been reported elsewhere (e.g Brown 1967, Smith et al. 2010; Romanovsky et al. 2010). As others have pointed out (see various papers already cited) the relationship is modulated by local factors. The incoming solar radiation that reaches the ground surface, and therefore influences the ground surface temperature and the deeper thermal regime, is probably the more important variable and probably not well captured in your data set.

**R:** We argue that here used estimate of potential incident solar radiation reaching the ground surface is not as important factor for MAGT as air temperature, because its effect in high latitudes is minimal for most of the year. Local topography also greatly affects the received amount of solar radiation at site. However, our results show that it still was more important (Fig. 3) and had a greater average effect (Table 2) than local soil and vegetation properties.

The amount of solar radiation is obviously a major control of climate in any area but conditioned regionally; Tibet plateau has high solar radiation but still low air temperatures, whereas e.g. the Nordic countries receive less solar radiation but are still mostly permafrost free. These anomalies clearly disrupt the northward decrease in MAGT and its association with solar radiation. We added a sentence to address the MAGT-solar radiation relationship (lines 259–261): *"Moreover, given that MAGT sites are usually located in more topographically heterogeneous terrain than ALT sites, the local exposure to solar radiation is suggested to be more important than the latitudinal trend (e.g. Romanovsky et al. 2010).*

**R2C28** Line 223-224 – Other studies (see those cited earlier) conclude that vegetation and soil properties are an important influence on the response of MAGT to changes in climate and therefore predictions of future conditions. Also, soil properties are an important influence on the thermal offset (which is not mentioned in this MS) which can be an important factor determining whether permafrost exists or not under warmer conditions (See previous comments).

**R:** We suppose the reviewer refers to lines 253-254. The effect of soil on MAGT was indeed small in our study but still not negligible. At this scale, it was anticipated that air temperature patterns would account for the greatest effect but the analyses were still able to point out their contribution, especially the effect size analysis, implicating that on average soil properties exerted a 0.4–0.7°C effect on MAGT. We modified the text to address these issues (lines 292–295): *"However, the effects of soil properties have been shown to be statistically significant when predicting future ground thermal conditions (Aalto et al., 2018), and should thus be considered. In addition, Throop et al. (2012), for example, concluded that substrate greatly affects the spatial distribution between permafrost, and that bedrock sites are expected to respond more rapidly to changes in climate than unconsolidated sediments."*

Concerning offsets, please see R2C3 and R2C18.

**R2C29** 271-272 – This has been concluded in other studies as mentioned in earlier comments.

**R:** We completely revised this sentence to acknowledge this has been concluded previously, and to focus on the new aspects of this study (lines 315–318): *"Our 1-km scale findings are congruent with previous process- and broad-scale studies stressing that, in addition to reliably addressing the key climatic factors, realistic modelling of Earth surface systems should take into account local-scale variation in solar radiation and ground properties."*

**R2C30** Line 273-275 – See earlier comments regarding importance of substrate conditions (soil or rock properties) in influencing the response of the ground thermal regime (and future permafrost conditions) to changes in climate.

**R:** We significantly modified the Conclusions to better reflect the findings of this study, and consequently removed this part (lines 318–320). This issue was discussed elsewhere (lines 292–295; R2C28).

observational dataset stressing that ALT strongly depends on local topo-edaphic factors (e.g. Harlan and Nixon, 1978) and that
thorough analyses of environmental factors controlling ALT at varying scales are urgently required.

Here, we use a statistical modelling framework employing multiple algorithms from regression to machine learning to examine
the  factors contributing to the spatial variation in the circumpolar ground thermal regime. More specifically,
we aim to (1) calibrate realistic models of MAGT and ALT (the responses) utilizing geospatial data on climatic and local
conditions (the predictors) across the Northern Hemisphere land areas, and (2)
examine the nature of the contributing factors in both permafrost and non-permafrost conditions using
 circumpolar field observations of MAGT and ALT.
 The analyses provide detailed insights into the importance of key environmental factors and the
magnitudes and direction of their effect at 1-km resolution.

**2 Methods**

**2.1 Study area and observational data**

We compiled MAGT and ALT observations from the period 2000–2014 over the Northern Hemisphere land areas north of the
30[th] parallel (Fig.1). To examine possible  differences in the contribution of environmental factors between permafrost
and non-permafrost conditions we used two separate MAGT datasets; observed MAGT at or below 0 °C, i.e. permafrost,
($MAGT_{\leq 0\ °C}$, n = 469) and above 0 °C ($MAGT_{>0\ °C}$, n = 315). For each MAGT and ALT site, averages over the study period
were then calculated from available annual averages or suitable single measurements. The observations were standardized by
requiring that MAGT was recorded at or near the depth of zero annual amplitude (ZAA) where annual temperature variation
was less than 0.1 °C, and that ALT (n = 298) values represented the maximum thaw depth of a given year based on mechanical

**Commented [A6]:** R1C10 Line 36: "ground temperatures are higher than air", adding "temperature" after "air", or "ground is warmer than air".

**Commented [A7]:** R2C9 Line 39-41 – Romanovsky et al. (2010) is probably a better reference to use here for the role of latent heat in determining the response of the ground thermal regime to changes in air temperature.

**Commented [A8]:** R2C6 Line 25-26 – One could argue that it is the presence of ground ice that influences the geomorphological processes and the impact of changing permafrost conditions.

**Commented [A9]:** R2C10 Line 46-47 – As mentioned above, there have been circumpolar and continental analyses of the environmental drivers.

**Commented [A10]:** R2C11 Line 52-54 – This observation wasn't unknown prior to Peng et al (2018) and as mentioned above, these relationships and the relevance of the "edaphic factors" are describe in variants of the Stefan Equation (e.g Harlan and Nixon, 1978; Nelson et al. 2000). Also, in an investigation of air temperature – ALT relationships across a range of ecoclimate zones, Smith et al. (2009) showed that the relationship varied according to vegetation and soil conditions (i.e. the edaphic factors).

**Commented [A11]:** R1C11 Line 59: "geographically comprehensive datasets of field-quantified MAGT (n = 784) and ALT (n = 298) observations.". Feels strange. How about "circumpolar field observations of MAGT and ALT". The number of sites in brackets can be described in methods section.

**Commented [A12]:** R2C1 This paper uses ground temperature and active layer data acquired from various sources to determine their relationships with various climate and other parameters to gain insights into the environmental drivers of the circumpolar ground thermal regime. While the analysis of this rather large data set is interesting (although others e.g. Peng et al. 2018 have made use of similar data sets) some of the insights regarding the various relationships are not necessarily new and have been reported elsewhere. In addition, some of the conclusions would appear to be at odds with those of other studies, which may be partly an issue of scale. A number of comments are offered below. These concerns should be addressed before the manuscript is considered for publication.

**Commented [A13]:** R1C12 Line 63: "possible variation …", not very clear/direct. Using "differences" instead of "variation".

**Commented [A14]:** R2C14 Line 60-83 – It is unclear whether the analysis utilizes a mean value for the entire 2010-2014 period for ground temperature, ALT, air temperature etc.

[revised manuscript text omitted]

**Commented [A15]:** R2C12 Line 67 – Was ALT only obtained through mechanical probing or were some values acquired through analysis of shallow ground temperature records. In the results section you give a maximum value of ALT of >7 m and it is unlikely that this was determined through mechanical probing. Some of the reports used for sources of ALT data may report ALT determined by methods other than probing (including thaw tubes and ground temperature measurements). Note also that probing does not necessarily capture the maximum thaw depth.

**Commented [A16]:** R2C13 Line 68 – The depth of ZAA can be much greater than 15 m and will depend on thermal properties of the subsurface materials. ZAA depth can be greater than 20 m for example in bedrock (see for eg. Romanovsky et al. 2010; Smith et al. 2010; Throop et al. 2012).

**Commented [A17]:** R1C13 Lines 71: "MAGT values shallower than two meters …" should be "MAGT measured at less than two meters …" . Delete "systematically".

**Commented [A18]:** R1C14 Line 85: "presenting", should be "representing"

**Commented [A19]:** R2C14 Line 60-83 – It is unclear whether the analysis utilizes a mean value for the entire 2010-2014 period for ground temperature, ALT, air temperature etc.

**Commented [A20]:** R1C15 Lines 94-98: You calculated TDD and FDD based on monthly climate data. Did you interpolated to daily or directly based on monthly averages? It is generally ok directly using monthly data based on the test of Frauenfeld et al. (2007. doi: 10.1002/joc.1372). You may refer to this paper for proof.

incident solar radiation, computed after McCune and Keon (2002, Equation 2, p. 605) utilizing slope angle and aspect, along with latitude, was used to estimate the potential incident solar radiation (PISR, W cm$^{-1}$ a$^{-1}$) that affects the energy balance of the ground thermal regime (e.g. Hasler et al., 2015; Streletskiy et al., 2015). Soil organic carbon content (SOC, g kg$^{-1}$), and fractions of coarse (CoarseSed, > 2 mm) and fine sediments (FineSed, ≤ 50 μm) for 0–200 cm subsurface, were extracted from SoilGrids database (Hengl et al., 2017).

**Commented [A21]:** **R1C21** What is the unit of solar radiation?

**2.3 Statistical modelling**

**2.3.1 Calibration of MAGT and ALT models**

We used four statistical techniques, namely generalized linear modelling (GLM, McCullagh and Nelder, 1989), generalized additive modelling (GAM, Hastie and Tibshirani, 1990), and regression-tree based machine-learning methods generalized boosting method (GBM, Friedman et al., 2000) and random forest (RF, Breiman 2001) to calibrate MAGT and ALT models by using the nine geospatial predictors. Multi-model framework was adopted to control for uncertainties related to the choice of modeling algorithm (e.g.  Marmion et al., 2009). GLM is an extension of linear regression capable of handling non-linear relationships with an adjustable link function between the response and explanatory variables. The GLM models were fitted including quadratic terms for each predictor. In GAM, alongside linear and polynomial terms, smoothing splines can be applied for more flexible handling of non-linear relationships. For smoothing spline, a maximum of three degrees of freedom were specified, which was further optimized by the model fitting function. To examine the direction and possible non-linearity of the relationship between predictors and responses, we used GAM to plot model-based  response curves. The curves show smoothed fit between response and a predictor while all other predictors are fixed at their average (Hjort and Luoto, 2011). Both GLM and GAM were fitted without interactions between predictors using a Gaussian error distribution with an identity link function.

**Commented [A22]:** **R1C20** Figure 4: I am not familiar with the GAM. It would be useful to briefly describe how the response shapes are calculated and what do they mean?

GBM was specified with the following parameters: number of trees = 3,000, interaction depth = 6, shrinkage = 0.001. Bagging fraction was set to 0.75 to select a random subset of 75 % of the observations at each step, without replacement. As for RF, 500 trees, each with a minimum node size of five were grown. The final prediction is the average of individual tree predictions. Both GBM and RF automatically consider interaction effects between predictors (Friedman et al., 2000). All statistical analyses were executed in R (R Core team, 2015) using auxiliary R packages; *mgcv* (Wood, 2011) for GAM, *dismo* (Hijmans et al., 2016) for GBM, and *randomForest* for RF (Liaw and Wiener, 2002).

**2.3.2 Model evaluation**

To evaluate the models, we split the response data randomly into calibration (70 % of the observations) and evaluation (30 %) datasets (Heikkinen et al., 2006). This was repeated 100 times, at each step fitting models with the calibration data and then using them to predict to both the calibration and evaluation datasets. Model performance was assessed with adjusted coefficient of determination ($R^2$) and root mean square error (RMSE) between observed and predicted values in these datasets.

**2.3.3 Variable importance computation**

A measure of variable importance was computed to determine the relative importance of each predictor to the models´ predictive performance (Breiman, 2001). In the computation, each modelling technique was first used to fit models with the MAGT and ALT datasets using all the nine predictors. The variable importance was then computed based on Pearson's correlation between predictions from two models produced with the fitted model; one with unchanged variables, and another where the values of one variable were randomized while others remained intact . In the procedure, each predictor was randomized in successive model runs. The measure of variable importance was computed as follows:

Variable importance = 1 − corr(Prediction$_{intact\ variables}$, Prediction $_{one\ variable\ randomized}$) (1)

On a range from 0 to 1, high variable importance value, i.e. high individual contribution to MAGT or ALT, was returned when any randomized predictor had a substantial impact on the model's predictive performance, and consequently resulted low correlation with predictions from the model with intact variables (Thuiller et al., 2009). Each modelling method was run 100 times for each response with each predictor shuffled separately. For each run, different subsample from the original data was randomly bootstrapped with replacement.

**2.3.4 Effect size statistics**

Effect sizes for each predictor were determined based on the range between the predicted minimum and maximum MAGT and ALT values over the observation data while controlling for the influence of other predictors by fixing them at their mean values (see Nakagawa and Cuthill, 2007). The procedure was repeated with each dataset and modelling method.

**3 Results**

MAGT in permafrost conditions was on average −3.1 °C while the minimum was −15.5 °C. MAGT$_{>0\ °C}$ had an average of 8.0 °C and a maximum of 23.2 °C. ALT had an average of 141 cm and ranged from 23 to 733 cm. The extreme values, apart from the ALT maximum, were based on one year of measurements. Pairwise correlations and the scatter plots revealed a strong association between MAGT and air temperature, especially in MAGT$_{>0\ °C}$ (Fig. 2a–b, d). In contrast to MAGT, ALT was not significantly correlated with TDD, but had stronger associations with soil properties (Fig. 2c). Coarse sediments and SOC, especially, were important and showed clear, yet non-linear, responses to ALT, respectively (Fig. 4c). Statistical descriptives of the predictors in respective datasets are presented  Fig. S1.

Commented [A23]: **R1C16** The tables and the figures are quite interesting. However, I feel the result section is a bit weak. I hope it can provide more detailed description, explanation and analysis of the tables and Figures.

**3.1 Model performance**

MAGT$_{>0\ °C}$ models had the highest $R^2$ values between predicted and observed MAGT (Table 1). In permafrost conditions, all the models had high $R^2$ values for MAGT, whereas in case of ALT between-model variation was large and $R^2$ on average lower. A decrease in the fit was identified when predicting ALT to evaluation datasets, especially with GBM and RF, whereas MAGT models retained their high performance. On average, RMSEs were low (~1 °C) in MAGT$_{<0\ °C}$ and MAGT$_{>0\ °C}$ calibration datasets. When predicted over evaluation datasets, the average increased slightly more in non-permafrost conditions. A similar increase of 40 % was documented with ALT. For each response, GBM and RF had lower RMSEs (i.e. higher predictive performance) than GLM and GAM, but also larger change  between calibration and evaluation datasets, indicating that GLM and GAM produced more  robust predictions.

Commented [A24]: **R1C16** The tables and the figures are quite interesting. However, I feel the result section is a bit weak. I hope it can provide more detailed description, explanation and analysis of the tables and Figures.

**3.2 Relative importance of individual variables**

FDD and TDD were the most important  factors affecting MAGT; FDD (0.27) where permafrost was present, TDD (0.53) in non-permafrost conditions (Fig. 3a–b). Precipitation predictors, especially water precipitation, had a moderate importance (0.10) on MAGT$_{≤0\ °C}$ but were marginal when permafrost was not present (0.01). Climatic  factors were followed by solar radiation (0.02, both MAGT datasets) and finally by NDVI and soil properties with minimal importance (each ≤0.01). The importance of both water and snow precipitation was higher in permafrost conditions.

Solar radiation was the most important predictor (0.37) explaining variation in ALT (Fig. 3c). Water precipitation had second highest importance (0.05) followed by soil properties SOC (0.04) and coarse sediments (0.03). The remaining climate variables (snow precipitation, TDD and FDD) had low importance scores that were comparable to those of NDVI (each 0.01–0.02).

**3.3 Effect size of individual variables**

FDD had the highest individual effect size of 6.7 °C averaged over the four methods in case of $MAGT_{\leq 0 °C}$, whereas in $MAGT_{>0 °C}$ dataset TDD accounted for a dominant 13.6 °C effect (Table 2). Precipitation had the  second highest effect, albeit snow precipitation was less effective in non-permafrost conditions. Considering the remaining predictors, clear differences were observed in cases of SOC and NDVI, both higher in $MAGT_{\leq 0 °C}$ dataset. In case of ALT, water precipitation exerted the greatest effect (181 cm) despite large between-model variation. In contrast to variable importance results (Fig. 3c), snow precipitation had a larger average effect than coarse sediments and SOC, both of which nevertheless had a considerable effect. Solar radiation had a central role with a highly non-linear shape of response (Fig. 4c). A varying degree of non-linearity is also visible in the responses between $MAGT_{\leq 0 °C}$ and the key predictors, whereas in case of $MAGT_{>0 °C}$ the responses are more linear (Fig. 4a–b).

**4 Discussion**

**4.1 Circumpolar  factors affecting  MAGT and ALT**

Our results  are in line with previous understanding that climatic conditions are the primary  factors affecting the long-term averages of circumpolar MAGT at 1-km resolution but also indicate that the effects of TDD and FDD on MAGT are dependent on  permafrost occurrence. As anticipated, FDD has higher influence on MAGT in permafrost conditions where strong freezing  is a prerequisite for the occurrence of permafrost (e.g. Smith & Riseborough, 1996). At sites without permafrost, TDD has the dominant nearly linear (Fig. 4b) effect, which is suggested to be mostly attributed to the lack of the buffering effect of the freeze-thaw processes and latent-heat exchange in the active layer (e.g. Osterkamp, 2007), and to the absence of seasonal  snow cover in the warmest parts of the study region. In permafrost conditions, the warming effect of TDD and especially the cooling effect of FDD on MAGT show flattening in response shapes where MAGT is close to 0 °C owing to the latent-heat effects associated with thawing and freezing of water in the active layer (Fig. 4a).

The minimal effect of TDD on ALT contradicts with the documented strong regional scale (spatio)temporal connection (e.g. Zhang et al., 1997; Oelke et al., 2003; Frauenfeld et al., 2004; Melnikov et al., 2004; Yi et al., 2018). According to our results, the spatial linkage is more elusive at a broader scale and could be attributed to the great circumpolar variation in ALT.  The majority of high-Arctic sites locate on low-lying tundra overlaid by mineral and organic soil layers, whereas at mid-latitudes (the Alps, central Asian mountain ranges) permafrost predominantly occurs in mountains with thin soils and thermally diffusive bedrock. This difference partly explains generally small and large ALT within the respective regions notwithstanding that they can have similar average climatic conditions (e.g. TDD, see Fig. 2d). Moreover, large inconsistencies between observed ALT and climate-warming trends have been documented (e.g. Wu et al., 2012; Gangodagamage et al., 2014). Although temporal dynamics of ALT are beyond our analyses, this suggests that thaw depth and air temperatures are, to a degree, decoupled by local conditions.

Recent warming trends in the atmosphere (Guo et al., 2017) are already well visible in circumpolar permafrost temperature observations (Romanovsky et al., 2017) implying that the permafrost system will remain dynamic in future's changing climate. Warmer air temperatures will occur mostly during winters (AMAP, 2017; Guo et al., 2017), which, given the presented high contribution of FDD on MAGT, suggests that changes are foreseeable. Projected warmer winters can also affect ALT through changing snow conditions and subsequent changes in hydrology and vegetation (Park et al., 2013; Atchley et al., 2016; Peng et al., 2018).

**Commented [A25]:** R1C16 The tables and the figures are quite interesting. However, I feel the result section is a bit weak. I hope it can provide more detailed description, explanation and analysis of the tables and Figures.

**Commented [A26]:** R2C20 Line 180-187 – I would disagree that this finding about the effects of TDD and FDD is all that significant. Cold conditions are a requirement for permafrost so FDDair will have a higher value in permafrost environments compared to non permafrost environments. This is described by the Frost Index model of Nelson and Outcalt (1983).

**Commented [A27]:** R2C21 Line 181 – Do you really mean there is a negative energy balance or do you mean FDD>TDD which is not the same thing (a negative energy balance would mean there is cooling over time).

**Commented [A28]:** R1C17 Line 184: This study did not include any sites with permanent snow cover, you may delete "or permanent snow cover".

**Commented [A29]:** R2C23 Line 189-193 – High Arctic sites do not necessarily have decimeter thaw depths. Greater thaw depths can be found in bedrock so the material type is important.

**Commented [A30]:** R2C24 As mentioned in previous comments, site specific factors are an important influence on ALT and its relationship with TDD and this is likely masked in your analysis.

**Commented [A31]:** R1C18 In discussion section, temporal changes were inferred from the spatial statistical results. The author should be cautious about that and clearly indicate the assumption.

According to Kurylyk et al. (2014), permafrost studies often consider only conductive heat propagation in the ground. Vincent et al. (2017), however, stress the need to acknowledge processes associated with liquid water and advective heat in efforts to understand rapidly changing cryosphere. In line with new studies (Peng et al., 2018; Zhang et al., 2018), our results highlight the notable role of water precipitation on both MAGT and ALT. Projected greater proportion of liquid precipitation (e.g. AMAP, 2017; Bintanja and Andry, 2017) potentially has a direct effect on the ground thermal regime through its influence on latent heat exchange (Westermann et al., 2011), and convective warming during spring (Kane et al., 2001) and summertime (Melnikov et al., 2004; Marmy et al., 2013). However, abundant summer rains arguably also cool the ground surface through increased evaporation and heat capacity, and thus limit the heat conduction into the ground (Zhang et al., 1997, 2005; Frauenfeld et al., 2004; Park et al., 2013). Moreover, extreme climatic events, such as wintertime rain events can have a distinct effect on soil temperature (Westermann et al., 2011) although the long-term sensitivity of permafrost to them is not fully clear yet (Marmy et al., 2013). According to Kurylyk et al. (2014), permafrost studies often consider only conductive heat propagation in the ground. Vincent et al. (2017), however, stress the need to acknowledge processes associated with liquid water and advective heat in efforts to understand rapidly changing cryosphere.

The dominant contribution of water precipitation over snowfall observed here contradicts with some previous regional scale studies (e.g., Zhang et al., 2003, 2005). However, the elevated effect of snowfall on MAGT in permafrost conditions (effect size of 2.3 °C compared to 0.8 °C in non-permafrost conditions) underlines the role of snow cover's control over the ground thermal regime. Similarly, Zhang et al. (2018) found that the offset between air and surface temperatures was weaker in temperate regions (mean annual air temperature >0 °C) than in low-Arctic and boreal permafrost regions, although also high-Arctic had small offsets owing to small amount of snow. Despite the complexity involved in snow's the role of snow conditions (e.g. Fiddes et al., 2015; Aalto et al., 2018b), thick snow cover has been shown to increase also ALT at site (Atchley et al., 2016), regional (Zhang et al., 1997; Frauenfeld et al., 2004) and circumpolar scale (Park et al., 2013).

Incoming solar energy can be considered central for soil thawing (see Biskaborn et al., 2015), but the high contribution of solar radiation on ALT stands out as well. Arguably, the effect is emphasized because ALT observation sites in cold permafrost conditions are mostly sparse in vegetation and lack tree canopy (Zhang et al., 2003; Biskaborn et al., 2015). Moreover, most of the ALT sites have been established on flat terrain (Biskaborn et al., 2015), meaning that local topographic shading is less significant. Thus, ALT is suggested to follow poleward decrease in solar radiation and associated shorter thaw seasons (see Luo et al., 2016). The weaker association of solar radiation with MAGT suggests that its direct effect is limited to the near-surface permafrost, i.e. intensified thawing during thawing seasons, and that the influence to deeper temperatures is more indirect and associated with the relationship between annual solar radiation and air temperatures. Moreover, given that MAGT sites are usually located in more topographically heterogeneous terrain than ALT sites, the local exposure to solar radiation is suggested to be more important than the latitudinal trend (e.g. Romanovsky et al. 2010).

The weak connection between TDD and ALT is additionally explained by soil factors that influence the heat transfer between the lower atmosphere and the ground (Smith et al., 2009). According to the response shapes from GAM, coarse sediments increase ALT when enough prevalent (~25 % fraction) in the soils. The effect of soil texture on ALT has been implied to occur largely through its effects on hydrological conditions (Zhang et al., 2003; Yin et al., 2017) and conductivity (Callaghan et al., 2011). More efficient water transfer in coarse-grained material could impose convective heat into soils during the thawing season or promote latent-heat effect during the freeze-up, which both contribute to deeper thaw (see Romanovsky and Osterkamp, 2000; Frauenfeld et al., 2004). Insulation by soil organic layers has been demonstrated to effectively decouple air-permafrost connection resulting in thinner active layer and lower soil temperatures (e.g. Johnson et al., 2013; Atchley et al., 2016). The GAM response shape illustrates a thinning of ALT with increasing SOC until ~150 g kg$^{-1}$, after which additional organic material does not attribute to enhanced insulation.

**Commented [A32]:** R2C25 Line 200-209 – The results presented don't really allow attribution of the effect of precipitation to advection over latent heat. Drainage will be an important factor.

**Commented [A33]:** R2C27 Line 218-225 – A general northward decrease in MAGT which is associated with decreasing solar radiation and air temperature has been reported elsewhere (e.g Brown 1967, Smith et al. 2010; Romanovsky et al. 2010). As others have pointed out (see various papers already cited) the relationship is modulated by local factors. The incoming solar radiation that reaches the ground surface, and therefore influences the ground surface temperature and the deeper thermal regime, is probably the more important variable and probably not well captured in your data set.

NDVI has a small contribution on ALT and MAGT in permafrost conditions, but outside the permafrost region it has a moderate cooling effect. The low contribution of NDVI in permafrost conditions could be attributed to the intra- and inter-seasonal differences in the effects of vegetation. In wintertime, low vegetation traps snow and thereby enhances insulation of the ground. Taller tree canopies of evergreen boreal forests, in turn, intercept snow and allow more heat loss from the ground in winter, while in summer their shading cools the ground surface (Lawrence and Swenson, 2011; Fisher et al., 2016).

**4.2 Uncertainties**

Large-scale scrutinization of factors affecting ground thermal dynamics is often hindered by data deficiencies or unavailability. More precisely, many data lack adequate spatial or temporal accuracy, geographical consistency, methodological robustness or thematic detail (Bartsch et al., 2016; Chadburn et al., 2017). Some of these shortcomings are exacerbated in remote permafrost regions with low-density observational networks of, e.g., climatic parameters (Hijmans et al., 2005) or soil profiles (Hengl et al., 2017). The fine-scale spatial variability of ALT and MAGT called for a high spatial resolution data to assess the local factors that mediate the atmospheric forcing. Here, the availability of geospatial data largely determined the resolution of 30 arc seconds, which could be considered the highest currently attainable resolution at a near-global scale. While not adequate to account for all potential sources of sub-grid spatial heterogeneity in, e.g. microclimatic conditions, especially in topographically complex conditions (Fiddes et al., 2015; Aalto et al., 2018b; Yi et al., 2018), the implemented resolution is a step forward in making a distinction in between-site conditions and revealing local relationships relevant at the circumpolar scale.

In general, the sensitivity of MAGT to the climatic parameters along with the minimal role of soil and vegetation properties suggests that circumpolar future predictions of MAGT are more applicable than those of ALT, even without addressing, for example, future vegetation or soil organic carbon content, whose response to climate change is extremely challenging to project (Jorgenson et al., 2013). However, the effects of soil properties on MAGT have been shown to be statistically significant when predicting future circumpolar ground thermal conditions (Aalto et al., 2018a), and should thus be considered. In addition, Throop et al. (2012), for example, concluded that substrate greatly affects the spatial distribution of permafrost, and that bedrock is expected to respond more rapidly to changes in climate than unconsolidated sediments. Given the pronounced role of precipitation, more direct information on fine-scale soil moisture conditions controlled by local soil and land surface properties (see Kemppinen et al., 2018), as well as more comprehensive and finer resolution data on circumpolar snow thickness are required for improved ground thermal regime modelling. Fine-scale biophysical factors affecting drainage conditions and distribution of wind-drifted snow (e.g. vegetation and small topo-graphic depressions) are largely averaged-out and cannot be accounted for at 1-km resolution.

Although the main  factors were identified as important and effective by each modelling technique, notable inter-modal variability suggested that using only one method could have led to disputable results. A multi-model approach was in this sense safer, although not all the methods may have worked optimally with the present observational and environmental data owing to their different abilities to handle collinearity, spatial autocorrelation or non-linearity. For example, interactions between variables were not included in regression-based modelling (GLM and GAM), while being intrinsically considered by tree-based methods (GBM and RF) (Friedman et al., 2000). Differences such as this could have attributed to the dissimilar performances of the models; GBM and RF were overall less stable when comparing $R^2$ and RMSE values between the observed and predicted values in calibration and evaluation settings.

**Commented [A34]:** R2C28 Line 223-224 – Other studies (see those cited earlier) conclude that vegetation and soil properties are an important influence on the response of MAGT to changes in climate and therefore predictions of future conditions. Also, soil properties are an important influence on the thermal offset (which is not mentioned in this MS) which can be an important factor determining whether permafrost exists or not under warmer conditions (See previous comments).

**Commented [A35]:** R2C8 Line 37-38 – One could argue it is the moisture content and drainage that are the important factors.

R2C15 Line 95-98 – Snow depth can be highly variable in northern environments depending on exposure to wind and vegetation. This is a site specific factor and its influence is probably not adequately considered by only utilizing precipitation records.

**5 Conclusions**

310 We assessed the factors affecting the circumpolar ground thermal regime at an unprecedentedly  high 1-km spatial resolution using comprehensive field-quantified observational datasets on MAGT and ALT. Our statistical modelling framework efficiently captured the multi-variate nature of ground thermal regime and highlighted the difference between the contributions of climatic factors on MAGT inside and outside the permafrost domain. In permafrost conditions, different key factors accounted for variation in MAGT and ALT; climate was paramount for MAGT, while local

315 environmental conditions were emphasized in case of ALT.  Our 1-km scale findings  are congruent with previous process- and broad-scale studies stressing that, in addition to reliably addressing the key climatic factors, realistic modelling  of Earth surface systems should take into account local-scale variation in solar radiation and ground properties.

320  In addition to providing theoretical insights about effective magnitudes and directions of the key contributing factors at circumpolar scale, multi-variate modelling frameworks capable of employing  high-resolution geospatial data  are valuable for the spatio-temporal prediction of ground thermal regime at circumpolar scale. .

> **Commented [A36]:** **R2C29** 271-272 – This has been concluded in other studies as mentioned in earlier comments.

> **Commented [A37]:** **R2C30** Line 273-275 – See earlier comments regarding importance of substrate conditions (soil or rock properties) in influencing the response of the ground thermal regime (and future permafrost conditions) to changes in climate.

> **Commented [A38]:** **R2C1** This paper uses ground temperature and active layer data acquired from various sources to determine their relationships with various climate and other parameters to gain insights into the environmental drivers of the circumpolar ground thermal regime. While the analysis of this rather large data set is interesting (although others e.g. Peng et al. 2018 have made use of similar data sets) some of the insights regarding the various relationships are not necessarily new and have been reported elsewhere. In addition, some of the conclusions would appear to be at odds with those of other studies, which may be partly an issue of scale. A number of comments are offered below. These concerns should be addressed before the manuscript is considered for publication.

[revised manuscript text omitted]

**Commented [A39]: R1C19** Figures 2a-c: one color legend probably is enough. Figure 2d: the units of TDD and FDD should be °C d.

[Figure]

**Figure 3: Variable importance values in MAGT$_{\leq 0\ °C}$ (mean annual ground temperature) (a) and MAGT$_{>0\ °C}$ (b) datasets arranged in the descending order of four-model average in MAGT$_{\leq 0\ °C}$ conditions, and for ALT (active-layer thickness) (c), arranged likewise based on ALT results. The whiskers depict 95 % confidence intervals (over 100 bootstrapping rounds). GLM = generalized linear modelling, GAM = generalized additive modelling, GBM = generalized boosting method and RF = random forest. See Sect. 2.2 for predictor abbreviations.**

540

[Figure]

**Figure 4: Response shapes of the five predictors with most contribution in MAGT$_{\leq 0\,°C}$ (a) (mean annual ground temperature, blue curves), MAGT$_{>0\,°C}$ (b) (red curves) and ALT (c) (active-layer thickness, yellow curves) datasets obtained from generalized additive modelling (GAM). Response shapes for the remaining predictors are illustrated in Figure S2. Predictors (see Sect. 2.2 for abbreviations) are presented in the descending order of their effect size in respective datasets. X-axis units appear in the original scale of the predictors. Y-axis displays partial residuals and labels the estimated degrees of freedom used in fitting the respective predictors to a response. Shaded areas depict 95 % confidence limits.**

545

---

## Author Response (AR2)

**Author response to editorial comments**

Line numbers refer to the marked-up manuscript at the end of this document.

Thank you for your revised manuscript. There are many improvements, but there is still room for improvement.

My major concern is that after reading the abstract, many will wonder what the new insights are. This echoes comments from Reviewers 1 and 2. You must highlight these insights in the abstract. In the manuscript you suggest that magnitude, direction and response shapes provide important insights, but little of this jumps out in the abstract. I fear that many readers will not read beyond this point in the paper. If the title is new insights, then you must deliver the goods.

The second concern, raised by Reviewer 2, is that some of your results are at odds with findings from much of the literature. This is not necessarily a bad thing, but you will strengthen your manuscript if you can elaborate on why you find these differences. If is is a matter of scale, then say so more explicitly. For example, if vegetation and soil conditions are not that important, is this simply because the integration of the analysis is at a global scale and vegetation and soil condition are more important at regional scales? Are there specific regions that strengthen or weaken the overall relations?

These concerns must be addressed before the manuscript can be considered for publication. I have suggested major revisions to give you time to address the comments.

**R:** We thank the Editor for the overall positive views and for pointing out places where we can improve our manuscript. We agree that the abstract should highlight the new findings better and recognized the need to comprehensively revise the abstract and the conclusion of this study. Moreover, we agree that contradictions with previous study results could be considered more clearly.

Detailed responses concerning discrepancies with previous studies, scale-related issues, and the regional aspects addressed by the Editor are provided below. In addition, we have carefully checked the manuscript for typos as well as made some updates to the references (added/removed a few references and included DOIs wherever applicable) and modified caption in Tables 1 and 2 as requested.

We hope that we have succeeded in addressing all the concerns and refining the manuscript to fully describe the performed study and explicitly discuss the relevant findings and associated caveats.

Line 2: This title is misleading given the number of sites well outside of polar regions. Do you really mean northern hemisphere?

**R:** We agree that using the Northern Hemisphere is more unambiguous. Revised accordingly: *"New insights into the environmental factors controlling the ground thermal regime in the Northern Hemisphere"*

Abstract: What are the new insights? The findings presented here largely confirm what is already known and will not surprise the reader, especially Lines16-19. Perhaps the conclusions here are too general? Provide some detail. E.g., what are the important non-linear influences from soil properties? Are there regional influences that contribute to the break down of relations that many would have expected to be stronger?

**R:** We agree that our conclusions may be cautious and thus too general. In the revised abstract, we emphasized findings that our novel (statistical techniques/high resolution data combined with hemispheric extent) approach yielded. In particular, we stressed the magnitudes (effect size in °C) and importance of key predictors for MAGT and ALT. In addition, shapes of responses are briefly summarized by addressing the

non-linearity found in responses between MAGT$_{\leq 0\,°C}$/ALT and environmental factors. Moreover, we included discussion on the influence of scale raised by the Editor and reviewer 2 as well as addressed the uncertainties associated with the used soil data. We feel that disclosing this already in the abstract is important when relating our study to the lineage of cryospheric science. Revised parts (lines 17–25):

*"Freezing (FDD) and thawing (TDD) degree-days were key factors for MAGT inside and outside the permafrost domain with average effect sizes of 6.7 °C and 13.6 °C, respectively. Soil properties had marginal effect on MAGT (effect size = 0.4–0.7 °C). For ALT rainfall (effect size = 181 cm) and solar radiation (161 cm) were most influential. Variable importance analysis further underlined the dominance of climate for MAGT and highlighted the role of solar radiation for ALT. Most response shapes for MAGT$_{\leq 0}$ °C and ALT were non-linear indicating thresholds for the covariation. It is suggested that the factors with large global variation (i.e. climate) suppressed the effect of local-scale factors (i.e. soil properties and vegetation) owing to the extensive study area and limited representation of soil organic matter. Our approach facilitates hemispheric-scale cryospheric studies by offering new insights into the factors affecting the ground thermal regime at a 1-km scale."*

Given that the modelling of the circumpolar ground thermal regime uses very high resolution climatic and local environmental factors, what can you now say about spatial variation of the relative importance of these factors? At a global scale we expect the findings presented here, but are there regions where SOC or vegetation are very important, and other regions where they are not at all important to cryospheric conditions? The approach taken glosses over regional differences that are important and might provide deeper insights than achieved (will likely also address many of the comments from Reviewer 2), and not discussing the spatial component of your data is a missed opportunity.

**R:** The Editor presents a very interesting idea for future studies. We assume that spatial variation in the relative contributions could exist. It is not straightforward to point out any specific regions, yet in the discussion we address the complexities in vegetation effects (different effects of trees and shrubs that cannot be explicitly separated with the used vegetation index) that may average out some relations (lines 286–291).

We are in a view that regional comparisons would require a new study with additional regional partitions as well as abandoning our present idea of "permafrost vs. non-permafrost" examination. The current study setting provided a great deal of results (e.g. we used four statistical techniques to determine the effect sizes and variable importance for nine environmental factors). A division of the study area to several sub-regions could provide deeper insights but would substantially complicate the presentation of the results and main message of the study. Consequently, we decided not to extend our study by introducing new sub-region based results.

Lines 44–45: Here you describe the surface offset. Pleas add in a sentence about the thermal offet.

**R:** A sentence was added in lines 44–45: *Soils have different heat conductivities between frozen or thawed states, which can result in notable temperature differences between ground surface and top of permafrost, i.e. thermal offset (e.g. Smith and Riseborough, 1996)."* We also elaborated that the other described offset is specifically surface offset (line 49).

Smith, M. W. and Riseborough, D. W.: Permafrost monitoring and detection of climate change, Permafrost and Periglac., 7, 301–309, 1996.

Line 49: Please just use rainfall or snowfall in the text, tables and figures.

**R:** Done.

Line 51: Where are there temperate soils in the circumpolar regions?

**R:** We notice the contradiction here and replaced *circumpolar* with *hemispheric*. Also elsewhere in the text, terminology was updated in this regard.

Affects what more directly?

**R:** Revised (line 53): *"…climate signal affects the ground thermal regime more directly…"*

Lines 69–75: Where do you find observations of ALT in non-permafrost conditions? Reviewer 2 had a related question (R2C5). Revise this statement and make it reflect your response to R2C5 so that it is more clear to the reader what you are trying to examine. Perhaps you should include an investigation of seasonal frost depth, or simplify things and keep within

**R:** The sentence here was undoubtedly a little ambiguous. Naturally, ALT observations were confined to permafrost regions. The part was revised to reflect that only MAGT observations were compiled outside permafrost regions. Simultaneously, we aimed at better addressing the issues raised by reviewer 2 by emphasizing a central aspect to our study: to examine how the factor contributions for MAGT differ between the two thermal regimes.

*"More specifically, we aim to (1) calibrate realistic models of the ground thermal conditions utilizing field observations of MAGT and ALT (the response variables) and geospatial data on climatic and local conditions (the predictors) across the Northern Hemisphere land areas, and (2) examine the relative importance, magnitude of effect, and response shapes of environmental factors at 1-km resolution. The focus of this study is on MAGT and ALT in permafrost regions but the analyses are also performed for sites with MAGT above 0 °C to compare factor importances, effect sizes and response shapes between the thermal regimes.*

The proposed investigation of seasonal frost depth is another interesting idea. However, we have not compiled data on freezing depth and are not aware of any suitable datasets at this extent. Regional studies exist; e.g., Zhang et al. (2003) and Frauenfeld et al. (2004, 2011) studied freeze depth with data from Russian meteorological stations, and Wang & Zhang (2014) and Peng et al. (2016) in NW China. Moreover, Oelke et al. (2003) performed a coarse-resolution simulation the depth of seasonally frozen ground over the entire Arctic drainage basin using reanalysis data.

Zhang, Y., Chen, W., and Cihlar, J.: A process-based model for quantifying the impact of climate change on permafrost thermal regimes, J. Geophys. Res., 108, D22, 4695, doi:10.1029/2002JD003354, 2003.
Frauenfeld, O. W., Zhang, T., and Barry, R. G.: Interdecadal changes in seasonal freeze and thaw depths in Russia, J. Geophys. Res., VOL. 109, D05101, doi:10.1029/2003JD004245, 2004.
Frauenfeld, O. W. and Zhang, T.: An observational 71-year history of seasonally frozen ground changes in the Eurasian high latitudes. Environ. Res. Lett., 6, 044024, 2011.
Wang Q.F. and Zhang T.: Spatiotemporal variations of maximum seasonal freeze depth in 1950s– 2007 over the Heihe River Basin, Northwest China, Sciences in Cold and Arid Regions, 6, 0209–0218, 2014.
Peng, X., Zhang, T., Cao, B., Wang, Q., Wang, K., Shao, W., and Guo, H.: Changes in freezing-thawing index and soil freeze depth over the Heihe River Basin, Western China, Arctic, Antarctic, and Alpine Research, 48:1, 161–176, 2016.
Oelke, C., Zhang, T., Serreze, M. C., and Armstrong, R. L.: Regional-scale modeling of soil freeze/thaw over the Arctic drainage basin, J. Geophys. Res., 108, D10, 4314, doi:10.1029/2002JD002722, 2003.

Line 92: corrected: meters → metres

Lines 129–130: The SoilGrids database reports soil organic carbon (fine earth fraction). What about soils with organics that have not substantially degraded (common inpermafrost peatlands)? Arguably, soils with the same SOC can have very different bulk densities/organic matter contents related to various stages of decomposition. Does SOC directly relate to Organic Matter Content? The two are not interchangeable. When we speak of the thermal offset and the role of the surface organic layer, we mean organic soil with a low bulk density, which is not the same as an organic rich soil with a high SOC. Is this adequately accounted for in your parameterization?

**R:** This is an important note, and we recognized the need to address the limitations of the SoilGrids SOC in the manuscript in a few places as described below.

We acknowledge that soil organic carbon content (SOC) does not equal soil organic matter content (SOM), and that thus SOC does not directly depict the amount of all organic material to which thermal offset is attributed. Moreover, we acknowledge that SOC and SOM are not interchangeable owing to properties of different soils although constant conversion factors between the two have been used (e.g. Nelson & Sommers 1996). However, suitable SOM data are not available, and physical fractionation of SOC is commonly used as a proxy owing to more straightforward measurement procedures (Bailey et al., 2017). In the development of SoilGrids data, SOM data (as well as total carbon observations) have been used to derive SOC using a conversion factor (Hengl et al. 2017).

Also included in the SoilGrids are SOC stock in tons per ha grids, which (in addition to SOC) employ bulk density and coarse fragments in their computation. Based on 10,000 randomly distributed sampling points over the study area (north of 30°N) the Spearman rank-order correlation (0.91, p<0.001) between SoilGrids SOC (g kg$^{-1}$) and SOC stock (t ha$^{-1}$) indicated that the used variable well depicts the spatial variability in density-considered SOC.

To address these issues we added a sentence in lines 281–285: *"It should be noted that the used variable depicts SOC in fine earth fraction and does not explicitly address incompletely decomposed or fresh organic matter, which are one of the central components of the thermal offset. However, suitable gridded data on soil organic matter content are not available, and physical fractionation of SOC has been commonly used as its correlative proxy owing to more straightforward measurement procedures (Bailey et al., 2017).*

Nelson, D. W. and Sommers, L. E.: Total Carbon, Organic Carbon, and Organic Matter, in: Methods of Soil Analysis. Part 3, Sparks, D. L., et al. (Eds.) Chemical Methods, SSSA Book Series No. 5, SSSA and ASA, Madison, WI, 961-1010, 1996.

Bailey, V. L., Bond-Lamberty, B., DeAngelis, K., Grandy, A. S., Hawkes, C. V., Heckman, K., Lajtha, K., Phillips, R. P., Sulman, B. N., Todd-Brown, K. E. O., and Wallenstein, M. D.: Soil carbon cycling proxies: understanding their critical role in predicting climate change feedbacks. Glob. Change Biol. 24, 895–905, 2017.

Hengl, T., Mendes de Jesus, J., Heuvelink, G.B.M., Ruiperez Gonzalez, M., Kilibarda, M., Blagotić, A., Shangguan, W., Wright, M. N., Geng, X., Bauer-Marschallinger, B., Antonio Guevara, M., Vargas, R., MacMillan, R. A., Batjes, N. H., Leenaars, J. G. B., Ribeiro, E., Wheeler, I., Mantel, S., and Kempen, B.: SoilGrids250m – Global gridded soil information based on machine learning, PLoS ONE 12, e0169748, doi.org/10.1371/journal.pone.0169748, 2017.

If not, is this a potential reason why your results place minimal importance on soil conditions despite the findings of so many others?

**R:** Data on organic layer thicknesses/organic matter content inarguably would have been important for the study. It is also possible that SoilGrids SOC stock variable could have addressed thermal offset better than the used SOC variable but given the high correlation between the two datasets(0.91, p<0.001) the use of

SOC stock variable would have probably not affected the main outcome of our study. We added a sentence in the Uncertainties section (lines 310–311): *"It is also possible, that the used SOC data could not fully address the thermal offset albeit ALT modelling showed realistic and moderately strong effects."*

At this extent and resolution the used predictor is to our opinion a suitable estimate of amount of organic material and occurrence of peatlands with high SOC/SOM. Moreover, derivation of SOC stock from modelled SOC, bulk density and coarse fragments incorporates uncertainties in each of its component (Tifafi et al., 2018). We separately took into account coarse fragments (indicative of low organic content when abundant).

Tifafi, M., Guenet, B., and Hatté, C.:Large differences in global and regional total soil carbon stock estimates based on SoilGrids, HWSD, and NCSCD: Intercomparison and evaluation based on field data from USA, England, Wales, and France. Glob. Biogeochem. Cycles, 32, 42–56, 2018.

Lines 179–181: Refer to others with similar results.

**R:** References added: *"Pairwise correlations and the scatter plots revealed a strong association between MAGT and air temperature (see Smith and Burgess, 2000; Smith and Riseborough, 2002; Throop et al., 2012), especially in MAGT>0 °C (Fig. 2a–b, d)."*

Smith, S. and Burgess, M.: Ground temperature database for Northern Canada, Geological Survey of Canada, Open File Report 3954, 2000.
Smith, M. W. and Riseborough, D. W.: Climate and the limits of permafrost: a zonal analysis, Permafrost and Periglac., 13, 1–15, 2002.
Throop, J., Lewkowicz, A. G., and Smith, S. L.: Climate and ground temperature relations at sites across the continuous and discontinuous permafrost zones, northern Canada, Can. J. Earth Sci., 49, 865–876, 2012.

Line 219: Amended as suggested: *current permafrost occurrence* → *permafrost presence or absence*

Line 220: Amended as suggested: *is a prerequisite for the occurrence of permafrost* → *occurs*

Line 221: Amended as suggested: *the dominant nearly linear* → *a nearly linear dominant*

Line 230–232: Amended as suggested: *"mid-latitude sites predominantly locate in mountains (the Alps, central Asian mountain ranges) with thin soils…"*

Lines 243,244 and 254: *water precipitation* → *rainfall.* Also elsewhere in suitable places.

Lines 278–281: This is where I wonder if SOC should not be equated to soil organic matter content or surface organic layer thickness. Please look carefully at how SOC is defined in the SoilGrids database, and how that relates to the thermal properties of the soil. How do you related g/kg to density? Organic soils with the most seasonal differences in thermal properties are those that are pure peat, 1000 g/kg.

**R:** In SoilGrids SOC data, the highest values ~600 g/kg are located in the extensive wetlands of W Siberian Lowlands indicating large amounts of organic material. That being said, we acknowledge that SOC density is higher in organic horizons than in mineral soil horizons of permafrost-affected soils. We cannot explicitly relate SOC values to density, and their use is thus an indirect estimate of thermal effect of organic soils, which we consider to be adequate.

To avoid equating SOC and SOM, we replaced *organic material* with *organic carbon* (line 280) and made an addition in lines (279–285): *"It should be noted that the used variable depicts SOC in fine earth fraction and does not explicitly address incompletely decomposed or fresh organic matter, which are one of the central components of the thermal offset. However, suitable gridded data on soil organic matter content are*

*not available, and physical fractionation of SOC has been commonly used as its correlative proxy owing to more straightforward measurement procedures (Bailey et al., 2017).*

Line 288: Amended as suggested: *vegetation → different vegetation canopy configurations.*

Lines 288–289: Reference added. Also, low vegetation was a little ambiguous, we now used tall shrubs *"In wintertime,  vegetation (e.g. tall shrubs) traps snow and thereby enhances insulation of the ground (Morse et al., 2012)"*

Morse, P. D., Burn, C. R., and Kokelj, S. V.: Influence of snow on near-surface ground temperatures in upland and alluvial environments of the outer Mackenzie Delta, Northwest Territories, Can. J. Earth Sci., 49, 895–913, 2012.

Line 305: Do you mean applicable or "appropriate".
"more applicable/appropriate than those of ALT" for what, or to what?

**R:** The text was revised (line 305): *"...suggests that  future  MAGT  is more  feasible to predict than  ALT, ..."*

Lines 311–315: As R2 suggests several times, many others have demonstrated the importance of soil soil and vegetation conditions. This paper contradicts those findings. Tell the reader what the new insight is, why soil and vegetation have minimal roles. What can you now tell us about the cryosphere that we didn't already know.

**R:** We added discussion about minimal contributions of soil properties and vegetation on MAGT (lines 306–311): *"This is incongruent with previous studies showing the high importance of soil properties for MAGT (e.g. Zhang et al., 2003; Throop et al., 2012). The discrepancies are argued to be partly attributed to the hemispheric study extent; large spatial variation in climatic parameters is suggested to have suppressed the effect of soil and vegetation properties locally. It is also possible that the used SOC data could not fully address the thermal offset albeit ALT modelling showed realistic and moderately strong effects."*

The low contribution of vegetation (NDVI) may stem from the reasons discussed in lines 286–291. The index is used to examine the amount of photosynthetic vegetation but it is unable to distinguish vegetation height, which generally defines whether vegetation traps snow (warming effect) or intercepts snow and shades the ground (tall canopies, cooling effect). However, MODIS NDVI has been shown to be closely related to fractional vegetation cover and has been used to obtain quantitative estimates of rainfall interception Galdos et al. (2012), and is thus suitable in estimating interception.

Please see the revised abstract and conclusions where we summarized the new insights more clearly than before.

Galdos, F. V., Álvarez, C., García, A., and Revilla, J. A.: Estimated distributed rainfall interception using a simple conceptual model and Model Resolution Imaging Spectroradiometer (MODIS), J Hydrol., 468–469, 213–228.

You need to explicitly discuss the thermal offset and how your approach treats it. the thermal offset is critical to MAGTs in discontinuous permafrost.

**R:** We agree that discussing offsets will position our work better in line with previous research. We introduced thermal offset in lines 44–45: *"Soils have different heat conductivities between frozen or thawed*

*states, which can result in notable temperature differences between ground surface and top of permafrost, i.e. thermal offset (e.g. Smith and Riseborough, 1996)."* In lines 128–131, we clarify the inclusion of soil properties: *"To account for the thermal offset dictated by soil properties (e.g., Smith and Riseborough, 1996, 2002; Kuryluk et al., 2014) we extracted soil organic carbon content … from SoilGrids database (Hengl et al., 2017)."*

In the discussion, we further described how the used soil properties were attributed to the thermal offset (lines 281–283): *"It should be noted that the used variable depicts SOC in fine earth fraction and does not explicitly address incompletely decomposed or fresh organic matter, which are one of the central components of the thermal offset."*

Kurylyk, B. L., MacQuarrie, K. T. B., and McKenzie, J. M.: Climate change impacts on groundwater and soil temperatures in cold and temperate regions: Implications, mathematical theory, and emerging simulation tools, Earth-Sci. Rev., 138, 313–334, 2014.

What about soil thickness? in SoilGrids, one property that is probably important is depth to bedrock. MAGT in bedrock with thin overburden will likely be very sensitive to climatic parameters. Much of the Canadian shield will be thus affected.

**R:** As discussed in lines 230–234, depth to bedrock is suggested to affect ALT through different thermal diffusivities of soils and bedrock. As the Editor points out, it also affects MAGT, and we briefly addressed this in lines 89–90 in the original manuscript.

However, we are not convinced about the suitability of SoilGrids depth-to-bedrock (DBT) variables in the present context. First, the censored DBT grids only consider bedrock surface (R horizon) found within 0-200 cm from surface, which is why values are heavily skewed with >90% of them being >200 cm (476 of 784 from all MAGT observations). Moreover, all the >200 cm values have been assigned a value of 200 making the variable problematic for examining statistical relationships. The absolute DBT variable, in turn, has an RMSE of > 8 metres, which to our opinion hinders the recognition of highly conductive consolidated materials in the critical first few metres from the surface.

In the early phases of the study, we tested a data on soil and sedimentary deposits thickness (Pelletier et al., 2016). This data shows DBT down to 50 m subsurface (hence encompassing all the used MAGT measurements). Owing to minimal variable importance and a flat response curve, we excluded it from further analyses. However, highly conductive soils were considered with the inclusion of the coarse sediments variable.

Pelletier, J. D., Broxton, P. D., Hazenberg, P., Zeng X., Troch, P. A., Niu, G.-Y., Williams, Z., Brunke, M. A., and Gochis, D.: A gridded global data set of soil, immobile regolith, and sedimentary deposit thicknesses for regional and global land surface modeling, J. Adv. Model. Earth Syst., 8, 41–65, 2016.

Is there a bias in your high-arctic/Lowland sites towards unconsolidated materials that should be noted?

**R:** Pelletier et al. (2016) also distinguished uplands and lowlands at 30 arc-sec resolution. We used these data layers to define lowland sites in the high-Arctic (Canadian Archipelago, Northern Greenland, Svalbard, Arctic Russian islands; as defined in AMAP 2011) and then calculated average sediment (unconsolidated material) thickness, also Pelletier et al. (2016).

- Out of 52 high-Arctic boreholes, 22 located in lowlands with an average sediment thickness of 22.1 m, 30 on uplands (0.5 m).
- For ALT, out of 13 high-Arctic sites 10 were in lowlands (19.8 m), 3 on uplands (0 m).

Lowland sites, (also by definition: "In lowlands, we refer to all unconsolidated material above bedrock as sedimentary deposits" (Pelletier et al., 2016)), have thicker layers of unconsolidated sediments than uplands. Considering the entire high-Arctic, MAGT sites seem not to be biased toward unconsolidated materials, ALT sites possibly, but sample is very small and data uncertainties may affect.

Chart below shows that 6 out of 22 high-Arctic lowland MAGT sites have <= 3 m sediment cover indicating that relatively thin covers are usual also in lowland sites. Whatsoever, these data cannot be used to determine whether there is bias toward unconsolidated sediments in favor of bedrock outcrops or peatlands, for example. We consider it very difficult to give an explicit answer to the Editor's comment, but consider it unlikely that the results are biased owing to the issue. Thus, we consider no need to mention this in the text.

[Figure]

[Figure]

AMAP, Arctic Climate Issues 2011: Changes in Arctic Snow, Water, Ice and Permafrost. SWIPA 2011
    Overview Report, 2012.

Lines 329–330: You really did the opposite of this. You didn't use the field data to tune up a 1-km spatial resolution model of the circumpolar ground thermal regime, instead you figured out the relative importance climate and environmental factors at the nearly 800 field sites using the 1-km spatially averaged data.

**R:** The revised sentence puts focus on the used methodology (lines 329–330): "*We statistically related observations of MAGT and ALT to high-resolution (~1 km2) geospatial data of climatic and local environmental conditions to explore the factors affecting the ground thermal regime across the Northern Hemisphere.*

Line 334: Not really circumpolar. More like northern hemisphere.

**R:** Amended accordingly here and elsewhere including the abstract.

Line 340–344: These are not strong new insights. These are well known and should not constitute a key element of your core conclusions. Please highlight the new insights that you have developed herein.

**R:** We agree on this and revised this part of the conclusions to emphasize the new findings of our study (lines 334–347): "*In permafrost conditions, different key factors accounted for variation in MAGT and ALT; climate was paramount and soil properties showed marginal role for MAGT, while local environmental conditions precipitation factors and topography-controlled solar radiation were emphasized in case offor ALT. Where permafrost was not present, precipitation was less influential and MAGT was predominantly controlled by air temperatures above 0 °C.*

*The relatively minor role of soil properties (especially organic carbon content) on MAGT and ALT may stem from the lack of global data with high local accuracy. The results also revealed distinct non-linear*

*relationships and thresholds between the ground thermal regime and environmental factors, especially in permafrost-affected regions. At sites without permafrost, responses were more often linear. In addition to providing  these insights about effective magnitudes and response shapes of the key contributing factors at hemispheric scale, it is concluded that multi-variate modelling frameworks capable of employing high-resolution geospatial data will be valuable for the spatio-temporal prediction of ground thermal regimes from local to global scale."*

**New insights into the environmental factors controlling the  ground thermal regime across the Northern Hemisphere**

Olli Karjalainen[1], Miska Luoto[2], Juha Aalto[2,3], and Jan Hjort[1]

[1]Geography Research Unit, University of Oulu, FI-90014, Oulu, Finland
5  [2]Department of Geosciences and Geography, University of Helsinki, FI-00014, Helsinki, Finland
[3]Finnish Meteorological Institute, FI-00101, Helsinki, Finland

*Correspondence to:* Olli Karjalainen (olli.karjalainen@oulu.fi)

**Abstract.** The thermal state of permafrost affects Earth surface systems and human activity in the Arctic and has
10  implications to global climate. Improved understanding of the local-scale variability in the  global ground
thermal regime is required to account for its sensitivity to changing climatic and geoecological conditions. Here, we
statistically related  observations of mean annual ground temperature (MAGT) and active-layer thickness (ALT)
to high-resolution (~1 km$^2$) geospatial data of climatic and local environmental conditions across the Northern Hemisphere.
The aim was to characterize the relative importance of key environmental factors and the magnitude and  shape of
15  their effects. The multivariate models fitted well to
MAGT and ALT observations with average R$^2$ values being ~0.94 and 0.78, respectively. Corresponding predictive
performances in terms of root mean square error were ~1.31 °C and 87 cm. Freezing (FDD) and thawing (TDD) degree-days
were key factors for MAGT inside and outside the permafrost domain with average effect sizes of 6.7 °C and 13.6 °C,
respectively. Soil properties had marginal effect on MAGT (effect size = 0.4–0.7 °C). For ALT rainfall (effect size = 181
20  cm) and solar radiation (161 cm) were most influential. Variable importance analysis further underlined the dominance of
climate for MAGT and highlighted the role of solar radiation for ALT. Most response shapes for MAGT$_{<0\,°C}$ and ALT were
non-linear indicating thresholds for the covariation. It is suggested that the factors with 
[revised manuscript text omitted]

345   insights about effective magnitudes and directions response shapes of the key contributing factors at circumpolar hemispheric scale, it is concluded that 
[revised manuscript text omitted]

---

## Author Response (AR3)

**Author response to editorial comments**

All line numbers refer to the marked-up manuscript at the end of this document.

As I read this manuscript again, I realized what was missing. The new insights largely stem from the response curves shown in Figure 4, but the figure and its meaning are treated in a cursory manner.

**What do these shapes tell us**? What is the **new understanding** / new insights? What are the **implications**? This analysis and discussion is what will really give your work some impact. The rest of the findings are largely in-line with others and are consequently of less interest, despite the robust modelling framework used to determine them.

I repeat this comment later on.

**R:** We sincerely thank the Editor for helping us to recognize the foremost potential of our manuscript. Our modelling framework produced a great deal of detailed outputs and we agree that the response shapes should be addressed with more attention. That being said, we think that variable importance and effect size analyses provided important information on the relative and absolute effects of the environmental controls, and thus we wish to retain also the previous results and discussion.

In this revision, we focused on explicitly communicating the results from the response shapes (Section 3 *Results*; lines 213–221); how they should be interpreted and what new information they provide (Section 4.1 *Factors affecting MAGT and ALT*; lines 232–234, 262–266, 272–274, 277–279 and 289–293). The advantages and limitations of our approach in assessing non-linearity were explained (4.2 *Uncertainties*; lines 356–358, also 264–266), followed by discussions on the implications of the findings (5 *Conclusions;* lines 375–379 and 380–387). All edits appear in the marked-up manuscript below and are detailed in the following comments. In addition, we made minor styling edits to the axis labels in Figs 2d, 4, S1 and S2, and column names in Table 2.

Title: This change will make the title relevant to The Cryosphere. Also, the study does not include sites that are not freezing-affected.

**R:** Thank you for the idea for a revised title. We agree that the title could be improved and made more relevant to The Cryosphere. However, we are afraid that the inclusion of "freezing-affected" could be slightly misleading. Owing to our comparative modelling setting, a small amount of the borehole sites was without seasonal frost. Thus, we suggest a revised title that reflects better our comparative data setting: *New insights into the environmental factors controlling the ground thermal regime across the Northern Hemisphere: a comparison between permafrost and non-permafrost areas.*

We consider that part of the novelty value of our study arises from the fact that we conducted the analyses inside and outside permafrost domain, although the main focus was still in the permafrost area and related processes (i.e. larger share of the boreholes were with $MAGT_{\leq 0\ °C}$; the included analyses of ALT; discussion focused on permafrost-environment relationships).

Line 20: Variable importance analysis → Analysis of variable importance

Line 22: Not defined so write out "MAGT that were <= 0 deg. C"

**R:** Done

Line 22: indicating → and indicated
Line 25: with → of

Line 27–28: Revised according to the suggestion (lines 27–28): *"Our  new insights into the factors affecting the ground thermal regime at a 1-km scale should improve future hemispheric-scale studies.*

Line 69: Need to keep a context relevant to the Cryosphere.

**R:** Edited to reflect the title (lines 69): *"...the hemispheric ground thermal regime in areas with and without permafrost."*

2.2 Predictor variables: Please add a line or two to explain why depth to bedrock was not used. I think that other readers will have the same question, and you explained why in your response.

**R:** We mentioned this where other soil properties were addressed (lines 125–128): *"Ground temperatures at sites with thin or no overlying unconsolidated sediments above bedrock have been shown to be more closely coupled with air temperatures than those with thick overburden and associated latent heat effects and lower thermal diffusivity (e.g. Throop et al., 2012). The effects of overburden thickness, however, could not be assessed due to the lack of suitable global fine-resolution data."*

Line 182: in → for

3 Results: You need a sub-section on the response shapes. The shapes are supposed to be an innovative contribution, yet Fig. 4 is really treated in a cursory manner. It should be treated with more detail.

**R:** We agree this is in place given the additional weight on the results from response curves. In the inserted sub-section in lines 213–221 the findings from key responses were summarized.

4 Discussion: Given that many new insights can come from the response shapes, please include a discussion. What are the implications of others ignore these non-linearities and treat them otherwise? This discussion will likely add a point of two to your conclusions and your abstract. These non-linearities and their meaning are perhaps the greatest contribution of the manuscript, and they are not well treated in the current version of the manuscript.

**R:** We thank the Editor for these recommendations and think that addressing those helped us better communicating our results.

To accompany the non-linearity addressed in the previous revision (lines 227–232 for the non-linear effect of air temperatures for $MAGT_{\leq 0 \,°C}$ (and linear for $MAGT_{>0 \,°C}$) and for the effects of coarse sediments and SOC for ALT in lines 295–296 and 302–303, respectively), we reflected additional responses central to the outcomes of this study. Several passages of text discussing the response curves were inserted in the Discussions (places indicated with line numbers below).

To start with, we better addressed the differing magnitude and shape of air temperature's effect on MAGT inside and outside permafrost domain (line 225). Next, we detailed the non-linear behavior of MAGT with air temperature, and also stressed the need to not neglect non-linearity in similar studies (lines 232–237). These findings were also reflected in the Conclusions (lines 375–376) and Abstract (lines 23–25).

We then added discussions on rainfall's response to MAGT and ALT with related uncertainty issues (lines 262–266), and then snowfall's response to $MAGT_{\leq 0 \,°C}$ (lines 272–274) and ALT (lines 277–279).

Solar radiation's response to MAGT and ALT was reflected in lines 289–293, and NDVI's response to MAGT was clarified to be linear in line 309.

Finally, we discussed uncertainties associated in the modelling methods and observation datasets, and suggested how assessing ALT-precipitation relation could be improved (lines 354–360).

4.1 Factors affecting MAGT and ALT: At some point in the discussion you should note that depth to bedrock (overburden thickness) likely affects MAGT, but that the Soil Grids DTB data were not suitable.

**R:** We agree on this and added a few sentences to the section in the Uncertainties section (lines 334–339):

"*Another soil property likely affecting MAGT and ALT is the thickness of overburden materials above bedrock. Suitable data, however, were not available to scrutinize this. For example, the depth to bedrock predictions in the SoilGrids data (Hengl et al., 2017) were not sufficient for assessing realistic responses because one of the measures considers the overburden thickness (depth to R horizon, i.e. intact regolith) only within the first two metres below surface. While another measure in the SoilGrids data covers the range of measured MAGT depths, it has an RMSE of over 800 cm making it simply too imprecise.*"

Line 310: Sentences revised per suggestions: *"...vegetation canopy configurations that can have similar index values. For example, In wintertime,  tall shrubs canopy traps snow and thereby enhances insulation of the ground (Morse et al., 2012), whereas taller tree canopies of evergreen boreal forests intercept snow and allow more heat loss from the ground in winter,  and in summer their shading cools the ground surface (Lawrence and Swenson, 2011; Fisher et al., 2016)."*

5 Conclusions: You will need to add in any additional points that relate to detailed discussion of the response shapes

Line 373: What do these shapes tell us? What is the new understanding / new insights? What are the implications? This what will really give your work some impact. The rest of the findings are largely in-line with others and are consequently of less interest, despite the robust modelling framework used to determine them.

**R:** We substantially improved this section by adding new conclusions related to the response shapes and communicating our results' and approach's implications.

First, we more explicitly concluded key results about the climate-permafrost relation (air temperature and precipitation response curves with $MAGT_{\leq 0\ °C}$) including interpretations of their implications in lines 375–379. These findings were now addressed also in the Abstract (lines 23–25).

The advantages and applicability of the used approach capable of addressing non-linearity were stressed in the revised sentences (lines 380–383). Lastly, we added some suggestions on how our modelling framework could be further used to facilitate model development and understanding of climate-permafrost relation (lines 383–387). These suggestions reflected the Editor's comments in the previous revision.

Line 381: it is concluded → we conclude

Line 392: You should probably acknowledge those who have given any helpful comments such as the referees.

**R:** We agree on this and added acknowledgments.

[revised manuscript text omitted]

---

## Author Response (AR4)

Dear Olli,

Thank you for your revisions. I think the manuscript is much improved. After a few minor edits the paper will be set to go.

All the best,

Peter

**R:** We sincerely thank the Editor for his valuable comments throughout the review process. The suggested edits made to the manuscript are detailed below along with the marked-up manuscript.

All line numbers refer to the corrected manuscript in a separate file.

Line 260: Revised as suggested: "… up to ~250 mm  above which…"

Line 271: *40 cm* → *400 mm*

Lines 273–275: Revised as suggested: "*Here,*  *active-layer* *thickening is visible only after relatively high snowfall values (~700 mm). However, this effect is based on a limited  set of ALT sites*  *(less than 10% of the ALT sites had snowfall exceeding 300 mm) and is therefore uncertain.*"

Line 370–371: Insertion made: "*...and that of snowfall started to level off at about 350 mm.*"

Lines 376–378: Revised as suggested: "*We suggest that comparable broad-scale assessments should be performed that would further discriminate continuous, discontinuous, and less extensive permafrost zones or geoecologically distinct regions* .*"

References:

We noticed a missing reference. Brown et al. (2002) was cited in the caption of Fig. 1 but not included in the References. This was now added in lines 434–435.

Table 2, Figures 2a–c, 3a–c and 4a–c. PISR is defined in Section 2.2, not SolarRad.

**R:** Great thanks for this careful notion. To make abbreviations consistent we replaced PISR with SolarRad in the section 2.2 (line 123). We considered SolarRad to be a more intuitive term to a reader than PISR, and thus chose to use it throughout the work.

Figure 1: Caption revised as suggested: "*The observational network of the  mean annual ground temperature (MAGT) and active-layer thickness (ALT) measurement sites across the Northern Hemisphere that were used in this study.*"

Subscripts were applied to $MAGT_{\leq 0\,°C}$ and $MAGT_{>0\,°C}$ in the figure legend.

Figure 2a–c: Panel titles were updated per suggestion: *(a)* → *(a) $MAGT_{\leq 0\,°C,}$ (b)* → *(b) $MAGT_{>0\,°C}$ and (c)* → *(c) ALT.*

Panel (d): Subscripts were applied to $MAGT_{\leq 0\,°C}$ and $MAGT_{>0\,°C}$ in the figure legend.

Figure 3a–c: Panel titles were updated per suggestion: *(a)* → *(a) $MAGT_{\leq 0\,°C,}$ (b)* → *(b) $MAGT_{>0\,°C}$ and (c)* → *(c) ALT.*

[revised manuscript text omitted]